# Fine pore engineering in a series of isoreticular metal-organic frameworks for efficient $C_2H_2$/$CO_2$ separation

Jun Wang[1], Yan Zhang[2], Yun Su[1], Xing Liu[1], Peixin Zhang[3], Rui-Biao Lin [4✉], Shixia Chen[1], Qiang Deng[1], Zheling Zeng[1], Shuguang Deng [5✉] & Banglin Chen [6✉]

The separation of $C_2H_2$/$CO_2$ is not only industrially important for acetylene purification but also scientifically challenging owing to their high similarities in physical properties and molecular sizes. Ultramicroporous metal-organic frameworks (MOFs) can exhibit a pore confinement effect to differentiate gas molecules of similar size. Herein, we report the fine-tuning of pore sizes in sub-nanometer scale on a series of isoreticular MOFs that can realize highly efficient $C_2H_2$/$CO_2$ separation. The subtle structural differences lead to remarkable adsorption performances enhancement. Among four MOF analogs, by integrating appropriate pore size and specific binding sites, $[Cu(dps)_2(SiF_6)]$ (SIFSIX-dps-Cu, SIFSIX = $SiF_6^{2-}$, dps = 4.4'-dipyridylsulfide, also termed as NCU-100) exhibits the highest $C_2H_2$ uptake capacity and $C_2H_2$/$CO_2$ selectivity. At room temperature, the pore space of SIFSIX-dps-Cu significantly inhibits $CO_2$ molecules but takes up a large amount of $C_2H_2$ (4.57 mmol g$^{-1}$), resulting in a high IAST selectivity of 1787 for $C_2H_2$/$CO_2$ separation. The multiple host-guest interactions for $C_2H_2$ in both inter- and intralayer cavities are further revealed by dispersion-corrected density functional theory and grand canonical Monte Carlo simulations. Dynamic breakthrough experiments show a clean $C_2H_2$/$CO_2$ separation with a high $C_2H_2$ working capacity of 2.48 mmol g$^{-1}$.

[1] School of Resource, Environmental and Chemical Engineering, Nanchang University, Nanchang 330031 Jiangxi, PR China. [2] Jiangxi University of Chinese Medicine, Nanchang 330031 Jiangxi, PR China. [3] Key Laboratory of Biomass Chemical Engineering of Ministry of Education, College of Chemical and Biological Engineering, Zhejiang University, Hangzhou 310027 Zhejiang, PR China. [4] MOE Key Laboratory of Bioinorganic and Synthetic Chemistry, School of Chemistry, Sun Yat-Sen University, Guangzhou 510006 Guangdong, China. [5] School for Engineering of Matter, Transport and Energy, Arizona State University, 551 E. Tyler Mall, Tempe, AZ 85287, USA. [6] Department of Chemistry, University of Texas at San Antonio, One UTSA Circle, San Antonio, TX 78249-0698, USA. ✉email: linruibiao@mail.sysu.edu.cn; shuguang.deng@asu.edu; banglin.chen@utsa.edu

Acetylene ($C_2H_2$) is a major raw feedstock for the production of various essential polymers and chemicals[1–3]. In industry, $C_2H_2$ is produced by partial $CH_4$ combustion and thermal hydrocarbon cracking, in which carbon dioxide ($CO_2$) is a worth-noting impurity that can show great impact upon the subsequent industrial processes[4,5]. Currently, energy-intensive solvent extraction and cryogenic distillations are employed to separate $C_2H_2/CO_2$ mixtures[6]. Due to the close boiling points (189.3 K for $C_2H_2$; 194.7 K for $CO_2$), these approaches suffer from low energy efficiency and are environmentally unfriendly[7–9]. Therefore, it is urgent to develop an energy-efficient approach to realize the challenging $C_2H_2/CO_2$ separation. Adsorption-based gas separation using porous materials represents a promising alternative technology[10–12]. Nevertheless, $C_2H_2$ and $CO_2$ gas molecules show identical molecular shapes (dimensions: $3.32 \times 3.34 \times 5.7$ Å$^3$ for $C_2H_2$; $3.18 \times 3.33 \times 5.36$ Å$^3$ for $CO_2$) and kinetic diameters (3.3 Å, Supplementary Fig. 1), making it very challenging to develop high-performance adsorbents for $C_2H_2/CO_2$ separation through physisorption[13–15].

Metal–organic frameworks (MOFs) are well-known for their readily tunable pore sizes/shapes and internal surface modification[16–19]. By virtue of the isoreticular principle and building blocks approach in MOF chemistry, the pore adjustment of porous materials has been performed in a more predictable and more precise way[20–24]. By substituting organic linkers and/or metal nodes, the pore space in MOFs can integrate shape matching and specific binding toward targeted gas molecules[25,26]. SIFSIX-type MOFs featuring anionic $MF_6^{2-}$ groups (M = Si, Ti, Ge, etc.) have been demonstrated as efficient adsorbents for many separations, mainly attributed to their high-density fluorinated sites and high-sieving pore[27]. The fluoride atoms can serve as hydrogen bonding acceptors forming strong interactions with $C_2H_2$[28]. On the other hand, the length of dipyridine linkers that defines the pore aperture is variable upon substitution, thus tuning the aperture size of one-dimensional (1D) pore channels in these MOF materials[29]. This uniqueness makes SIFSIX-type MOFs a prominent platform with several progresses for separation such as $C_2H_2/C_2H_4$[30], $C_3H_4/C_3H_6$[31], C4 isomers separation[32], and $CO_2$ capture[33]. Among these SIFSIX-type materials, a flexible MOF $[Zn(dps)_2(SiF_6)]$ (UTSA-300-Zn, SIFSIX-dps-Zn) was able to completely differentiate $C_2H_2$ and $CO_2$ gas molecules. However, flexible MOFs usually show negligible gas uptake before gate-opening, which might lead to capture leakage when applied to the breakthrough separation of gas mixture[34–38]. In this context, flexible-robust MOFs with permanent small pores as well as specific binding sites have been employed to selective take up targeted gas molecules, whereas minimizing the co-adsorption of counterpart gases by tuning the gate-opening pressure, which has been demonstrated by $[Cu(dps)_2(SiF_6)]$ (SIFSIX-dps-Cu) for size-exclusive adsorption of $C_2H_2$ from $C_2H_4$[39]. To achieve simultaneously high capacity and separation selectivity for more challenging $C_2H_2/CO_2$ separation, a systematic study on fine-tuning of pore structure in prototypal $[Zn(dps)_2(SiF_6)]$ would be a rational approach.

Herein, we demonstrate precise control over pore structure through altering anionic linkers and metal nodes in $[Zn(dps)_2(SiF_6)]$ to increase both $C_2H_2$ uptake capacity and $C_2H_2/CO_2$ selectivity. Combining different anionic linkers, three flexible-robust MOFs with precise modulation of pore cavity sizes in a sub-nanometer scale have been utilized for the challenging $C_2H_2/CO_2$ separation. By integrating suitable pore size and fluorinated binding sites, the exclusive $C_2H_2$ sorption behavior was retained in $[Cu(dps)_2(SiF_6)]$ (SIFSIX-dps-Cu, SIFSIX = $SiF_6^{2-}$, dps = 4.4′-dipyridylsulfide, also termed as NCU-100) with a $C_2H_2$ uptake of 4.57 mmol g$^{-1}$ and negligible $CO_2$ uptake, resulting in a high IAST selectivity of 1787 for $C_2H_2/CO_2$

separation. The highly efficient $C_2H_2/CO_2$ separation in $[Cu(dps)_2(SiF_6)]$ has been validated by molecular modeling studies and dynamic breakthrough experiments.

## Results

**Synthesis and characterization.** A series of SIFSIX-dps-Zn variants, SIFSIX-dps-Cu (SIFSIX = $SiF_6^{2-}$, dps = 4.4′-dipyridylsulfide, termed as NCU-100), GeFSIX-dps-Cu (GeFSIX = $GeF_6^{2-}$), and NbOFFIVE-dps-Cu (NbOFFIVE = $NbOF_5^{2-}$) were successfully prepared through solution reactions (Fig. 1a, see Supplementary Information for synthetic and crystallographic details). Crystal structures of the as-synthesized MOFs were determined by single-crystal X-ray diffraction studies, and the phase purities of as-synthesized and activated samples were confirmed by the XRD measurements (Supplementary Figs. 2–4 and Supplementary Table 1). Each Cu(II) atom connects four independent pyridinyl rings of dps ligands and affords 1D chains, generating the intralayer cavity (Site I) with the size of $3.0 \times 3.2$ Å$^2$, $2.5 \times 3.1$ Å$^2$, and $2.3 \times 3.1$ Å$^2$ on NbOFFIVE-dps-Cu, GeFSIX-Cu-dps-Cu, and SIFSIX-dps-Cu, respectively (Fig. 1b and Supplementary Fig. 5a). These apertures are larger than those of the intralayer channels ($2.2 \times 3.1$ Å$^2$) in UTSA-300 (SIFSIX-dps-Zn). The chains are further bridged by different anion pillars in the perpendicular direction at Cu(II) sites to form 2D MOF layers containing 1D wavy interlayer channels. The 2D MOF layer planes stack with each other via multiple hydrogen bonds between guest water molecules and F atoms of anion pillars, rendering the structural flexibility and dynamics (Supplementary Fig. 6). The size of the interlayer cavity (Site II) on NbOFFIVE-dps-Cu, GeFSIX-Cu-dps-Cu, and SIFSIX-dps-Cu are $3.2 \times 4.8$ Å$^2$, $2.8 \times 4.8$ Å$^2$, and $2.9 \times 4.4$ Å$^2$, respectively (Fig. 1c and Supplementary Fig. 5b). These results illustrate that both interlayer and intralayer cavities can be finely tuned by altering the anion pillars due to different M–F distances (1.69 Å for Si···F, 1.75 Å for Ge···F, and 1.81 Å for Nb···F).

Although the as-synthesized samples have a similar layered stacking pattern, notable structural transformations are observed after activation, and the activated crystal structures are determined by Rietveld refinements (Fig. 1 and Supplementary Fig. 7 and Supplementary Tables 2–5). In contrast to the Zn analog UTSA-300 (Site I: $1.3 \times 2.8$ Å$^2$ vs. $2.3 \times 3.1$ Å$^2$ for as-synthesized), all Cu-based dynamic layered MOFs displayed slightly expanded intralayer cavity owing to the elongated Cu–F bonds (2.27 Å) than Zn–F bond (2.09 Å). As shown in Fig. 1, the activated NbOFFIVE-dps-Cu showed the largest pore aperture size of $2.2 \times 2.7$ Å$^2$ at Site I, in contrast to those in GeFSIX-dps-Cu ($1.5 \times 3.0$ Å$^2$) and SIFSIX-dps-Cu ($1.4 \times 3.0$ Å$^2$). The expanded pore spaces are conducive for potential $C_2H_2$ diffusion. Furthermore, the removal of solvent allows the interlayer π–π stacking of dps ligands, giving closed interlayer cavities (Supplementary Fig. 8). The interlayer distance (S atoms to the 2D layer center) was measured to be 4.10 Å in SIFSIX-dps-Cu, and followed by GeFSIX-dps-Cu (4.06 Å), UTSA-300 (4.05 Å), and NbOFFIVE-dps-Cu (3.69 Å), leading to varying interlayer pore spaces upon different packing density[13,40]. The changes of powder X-ray diffraction (PXRD) patterns and corresponding structural transformation in the three isoreticular MOFs upon desolvation or $C_2H_2$-loading are the same as the prototypical zinc analogue UTSA-300 (Supplementary Figs. 2–4 and Supplementary Tables 2–5)[13].

**Adsorption and separation performances.** The permanent porosity of these dynamic layered MOFs is probed by $CO_2$ adsorption isotherms at 195 K, and the Brunauer–Emmett–Teller specific surface area was determined as 358, 310, and 173 m$^2$ g$^{-1}$ for SIFSIX-dps-Cu, GeFSIX-dps-Cu, and NbOFFIVE-dps-Cu,

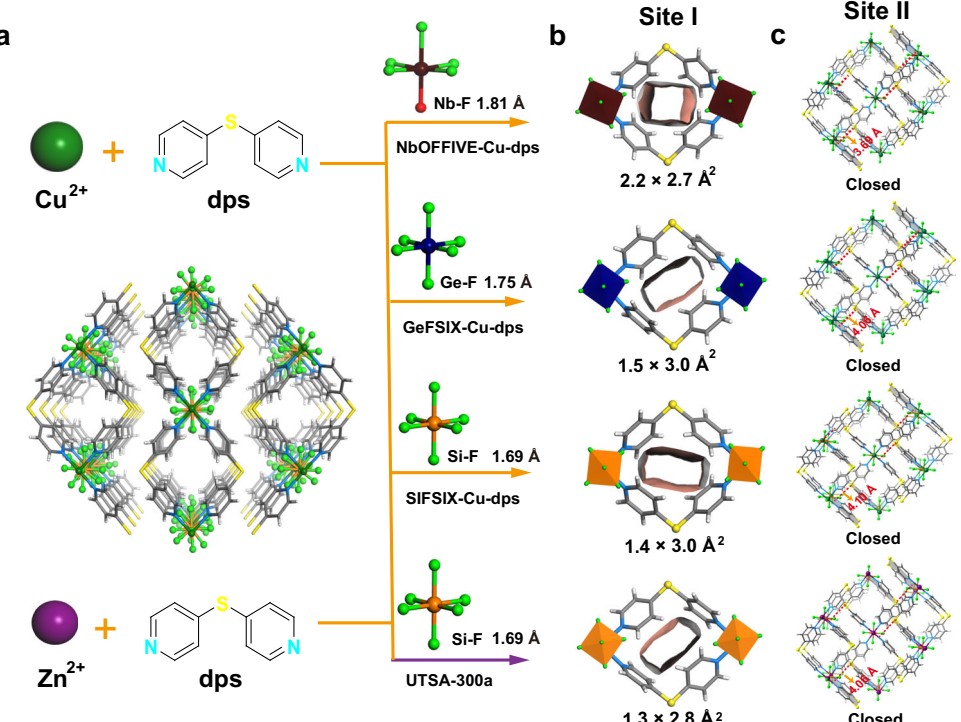

**Fig. 1 Crystallographic structures. a** Synthesis procedure of three isoreticular MOFs. Views of **b** Site I and **c** Site II in activated NbOFFIVE-dps-Cu, GeFSIX-dps-Cu, SIFSIX-dps-Cu, and UTSA-300a with varying pore aperture size. Color code: Cu, green; F, light green; S, bright yellow; N, light blue; C, gray; Si, orange; Ge, navy blue; Nb, wine red; and solvent molecules are omitted for clarity.

respectively (Supplementary Figs. 9 and 10). It should be noted that NbOFFIVE-dps-Cu exhibits stepwise adsorption behavior with a smaller pore volume (0.09 cm$^3$ g$^{-1}$, at $P/P_0$ ~0.25) before gate-opening as compared to those of SIFSIX-dps-Cu (0.20 cm$^3$ g$^{-1}$) and GeFSIX-dps-Cu (0.19 cm$^3$ g$^{-1}$), although it shows a potential total pore volume of 0.30 cm$^3$ g$^{-1}$. Thermogravimetric analysis revealed that these layered MOFs are stable up to 443 K (Supplementary Fig. 11). The XRD patterns showed that the activated structures can be restored to the as-synthesized state after being placed in the air for 24 h (Supplementary Figs. 2–4). Single-component equilibrium adsorption isotherms of C$_2$H$_2$ and CO$_2$ were collected at 273, 298, and 323 K (Supplementary Figs. 12–14). The C$_2$H$_2$ uptake of SIFSIX-dps-Cu, GeFSIX-dps-Cu, and NbOFFIVE-dps-Cu was measured to be 4.57, 4.04, and 1.65 mmol g$^{-1}$ at 298 K and 1.0 bar, respectively (Fig. 2a). The high C$_2$H$_2$ uptake of SIFSIX-dps-Cu (4.57 mmol g$^{-1}$) outperforms many benchmark MOFs, including CPL-1-NH$_2$ (1.84 mmol g$^{-1}$)[41], NTU-65 (3.36 mmol g$^{-1}$)[34], and JNU-1 (2.1 mmol g$^{-1}$)[5], see Supplementary Table 6. The sorption behaviors of these dynamic MOFs are similar to those in relevant literature[39,42]. The three MOF show similar C$_2$H$_2$ adsorption capacities (~0.85 mmol g$^{-1}$) at the low-pressure region (Fig. 2b). This gate-opening phenomenon can be attributed by structural dynamics including the rotation of pyridinyl rings upon C$_2$H$_2$-loading. There are differences on the torsion angle between anion pillar and dps ligand before and after C$_2$H$_2$ loading in SIFSIX-dps-Cu (7°), GeFSIX-dps-Cu (6°), and NbOFFIVE-dps-Cu (1°), all of which are much smaller than that of UTSA-300a (17°) (Supplementary Figs. 15–16), matching well with corresponding static sorption results. The C$_2$H$_2$ threshold pressure for gate-opening of NbOFFIVE-dps-Cu is around 0.3 bar at 273 K, which was significantly higher than those of 0.06, 0.05, and 0.035 bar for UTSA-300a, GeFSIX-dps-Cu, and SIFSIX-dps-

Cu, respectively (Fig. 2c). The dynamic adsorption behaviors were consistent with the trend of interlayer π–π stacking distances.

These results demonstrate that the subtle change of dynamic layered MOFs by substituting different pillared anions can greatly impact the C$_2$H$_2$ adsorption behavior, which would ultimately affect the selectivity of C$_2$H$_2$/CO$_2$. At 298 K, NbOFFIVE-dps-Cu showed smooth CO$_2$ adsorption of 1.10 mmol g$^{-1}$. In contrast, GeFSIX-dps-Cu and SIFSIX-dps-Cu show minor CO$_2$ uptake at low-pressure regions, although the latter shows stepwise CO$_2$ uptake at 273 K and pressures above 0.7 bar (2.34 mmol g$^{-1}$ at 1.0 bar, Fig. 2d). This might be ascribed to the gate-opening effect in the relatively flexible framework of SIFSIX-dps-Cu upon strong interactions with CO$_2$. To investigate the adsorption phenomena for gas mixture, mixed-components adsorption isotherms of the three isoreticular MOFs for C$_2$H$_2$/CO$_2$ (50/50, mol/mol) have also been collected (Supplementary Fig. 17). Compared with the single component sorption results, their adsorption capacity and threshold pressure are basically not changed, which confirms the preferential adsorption of C$_2$H$_2$ from C$_2$H$_2$/CO$_2$ mixture[43]. Prompted by the dramatic uptake differences, the ideal adsorbed solution theory (IAST) was applied to qualitatively estimate the C$_2$H$_2$/CO$_2$ selectivity, while the adsorption isotherms are fitted by the dual-site Langmuir–Freundlich equation with excellent accuracy (Supplementary Figs. 18–29 and Supplementary Table 7). As shown in Fig. 2e and Supplementary Fig. 30, the calculated equimolar C$_2$H$_2$/CO$_2$ selectivity of NbOFFIVE-dps-Cu, GeFSIX-dps-Cu, and SIFSIX-dps-Cu at 298 K and 1.0 bar are 9, 172, and 1787, respectively. In particular, the selectivity of SIFSIX-dps-Cu is higher than that of UTSA-300a (743) and much higher than other benchmark MOFs (Supplementary Table 6), such as CPL-1-NH$_2$ (119)[41], ATC-Cu (53.6)[9], TIFSIX-2-Cu-i (6.5)[44], UTSA-74a (9)[45], and UTSA-222a (2)[46]. Compared with those of adsorbents with high C$_2$H$_2$ adsorption capacities such as

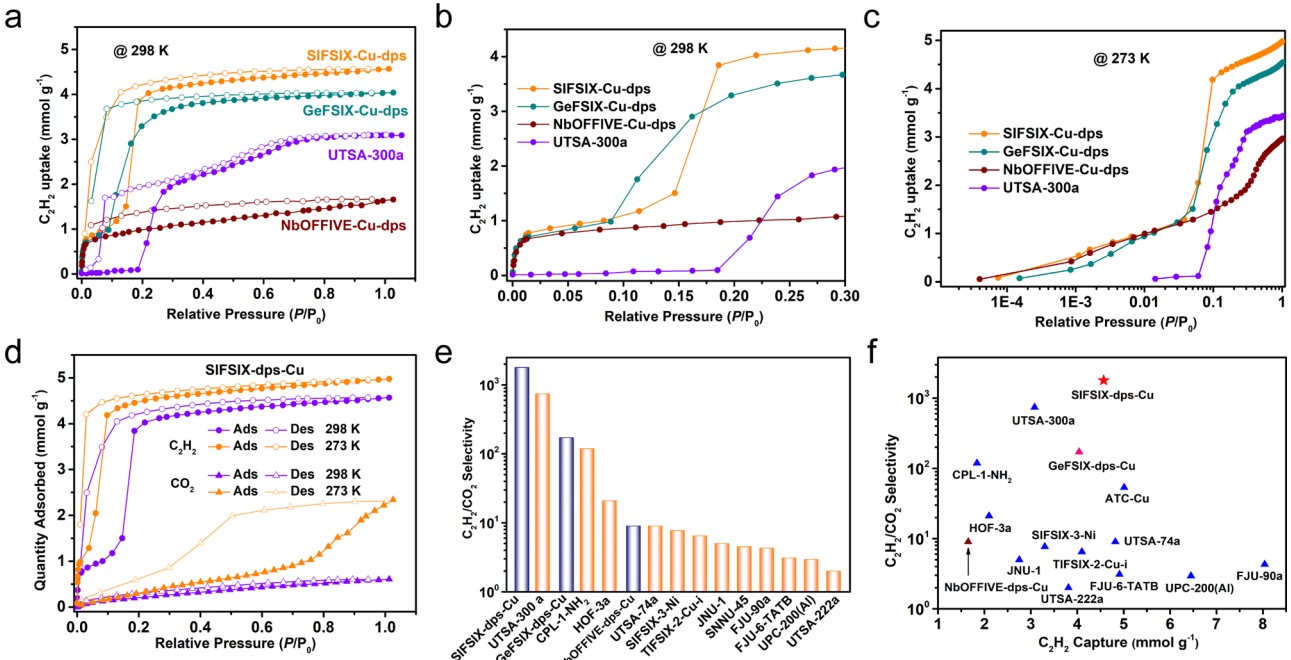

**Fig. 2 C$_2$H$_2$ and CO$_2$ sorption in four isoreticular MOFs.** The C$_2$H$_2$ sorption isotherms at **a** 0–1.0 bar and **b** 0–0.3 bar under 298 K. **c** The C$_2$H$_2$ sorption isotherms at logarithmic 0.01–1.0 bar of SIFSIX-dps-Cu, GeFSIX-dps-Cu, NbOFFIVE-dps-Cu, and UTSA-300a at 273 K. **d** the C$_2$H$_2$ and CO$_2$ adsorption isotherms of SIFSIX-dps-Cu at 273 and 298 K. **e** comparison of IAST selectivity of representative MOFs for 50/50 C$_2$H$_2$/CO$_2$. **f** Comparison about C$_2$H$_2$/CO$_2$ selectivity and C$_2$H$_2$ capacity of representative MOFs at 298 K and 1 bar.

SNNU-45[47], FJU-90a[48], UPC-200(Al)[49], and FJU-6-TATB[50] (Fig. 2f), SIFSIX-dps-Cu and GeFSIX-dps-Cu are still outperforming.

**Modeling simulation studies.** To investigate the potential C$_2$H$_2$ adsorption sites in these layered MOFs, dispersion-corrected density functional theory (DFT-D) and grand canonical Monte Carlo (GCMC) simulations are further carried out. Given that it is difficult to get the structures of intermediate states during the dynamic adsorption whereas the structural change during C$_2$H$_2$ loading is similar to UTSA-300, the activated or open frameworks were thus used for simulations. The distribution density of C$_2$H$_2$ was investigated firstly at 1 kPa, as shown in Fig. 3a, b and Supplementary Figs. 31–33, only C$_2$H$_2$ can be adsorbed in intralayer cavities (Site I) on all for isoreticular MOFs. As the loading pressure increased to 100 kPa, the interlayer cavities (Site II) became accessible by C$_2$H$_2$ stimuli (Fig. 3b). Such adsorption behavior is in line with that in UTSA-300 confirmed by neutron diffraction data[13]. In contrast, there is no gate-opening sorption for CO$_2$ even at 100 kPa that may be attributed to the opposite molecular quadrupole moment (−13.4 × 10$^{−40}$ C m$^2$ for CO$_2$ and +20.5 × 10$^{−40}$ C m$^2$ for C$_2$H$_2$)[13,51]. The C$_2$H$_2$ uptake of SIFSIX-dps-Cu, GeFSIX-dps-Cu, and NbOFIVE-dps-Cu was also evaluated by GCMC simulation showing the capacity of 4.40, 3.71, and 1.48 mmol g$^{−1}$, respectively, which are comparable to their experimental uptakes. Moreover, DFT-D calculations provide insight into the adsorption behaviors and the calculation details are provided in the "Methods" section. The adsorption sites are similar in three dynamic layered MOFs, with C$_2$H$_2$ molecule bonded by four F atoms of two distinct fluorinated anion pillars at Site I and Site II via cooperative H···F hydrogen-bond interactions (1.85–2.36 Å, Fig. 3c and Supplementary Fig. 34). The static binding energy of SIFSIX-dps-Cu for C$_2$H$_2$ is calculated to be 60.3 and 67.5 kJ mol$^{−1}$, respectively, which are also higher than those in GeFSIX-dps-Cu (58.5 kJ mol$^{−1}$) and NbOFIVE-dps-Cu (55.3 kJ mol$^{−1}$). The abundant binding sites with high

static binding energies are responsible for the outstanding C$_2$H$_2$ uptake of SIFSIX-dps-Cu (Fig. 3d). In contrast, CO$_2$ in SIFSIX-dps-Cu interacts with pore surface through weak intermolecular interactions like electrostatic interactions (F$^{δ−}$···C$^{δ+}$ 2.91–3.63 Å, Supplementary Fig. 35). The change in torsion angle between anion pillar and dps ligand caused by CO$_2$ loading is about 18°, which is larger than that for C$_2$H$_2$ loading (7°), in line with higher gate-opening pressure for CO$_2$ sorption. The experimental isosteric adsorption enthalpy ($Q_{st}$) at zero-coverage for C$_2$H$_2$ in SIFSIX-dps-Cu is 60.5 kJ mol$^{−1}$ (Supplementary Fig. 36), slightly higher than the $Q_{st}$ of GeFSIX-dps-Cu (56.3 kJ mol$^{−1}$) and NbOFIVE-dps-Cu (53.6 kJ mol$^{−1}$).

**Transient breakthrough experiments.** To confirm the practical C$_2$H$_2$/CO$_2$ separation performances of these isoreticular MOFs, experimental breakthrough experiments were conducted with SIFSIX-dps-Cu, GeFSIX-dps-Cu, and NbOFIVE-dps-Cu at 298 K. Binary mixture (C$_2$H$_2$/CO$_2$, 50/50, v/v) were injected into a packed column with a flow rate of 2.0 ml min$^{−1}$ and the clean separations of C$_2$H$_2$/CO$_2$ mixtures were realized by all dynamic layered materials (Supplementary Figs. 37 and 38). As expected, SIFSIX-dps-Cu exhibits the best C$_2$H$_2$/CO$_2$ separation performance. In Fig. 4a, CO$_2$ broke through the bed quickly after feeding the gas mixture into the fixed adsorption column, whereas the C$_2$H$_2$ was retained in the adsorption bed for 53 min g$^{−1}$. This value is comparable to GeFSIX-dps-Cu (50 min g$^{−1}$) and significantly outperforms NbOFIVE-dps-Cu (14 min g$^{−1}$) and UTSA-300a (12 min g$^{−1}$) at similar conditions. For small roll-up of the breakthrough curves for CO$_2$ in SIFSIX-dps-Cu, it can be attributed to the desorption of CO$_2$ induced by C$_2$H$_2$, which indicates a minor co-adsorption of CO$_2$ during the dynamic capture process and finally displaced by C$_2$H$_2$. That phenomenon is consistent with the minor CO$_2$ uptake from single-component sorption isotherms. The C$_2$H$_2$ and CO$_2$ adsorption kinetics in SIFSIX-dps-Cu showed no significant CO$_2$ uptake under a sufficiently long period of time but rapid C$_2$H$_2$

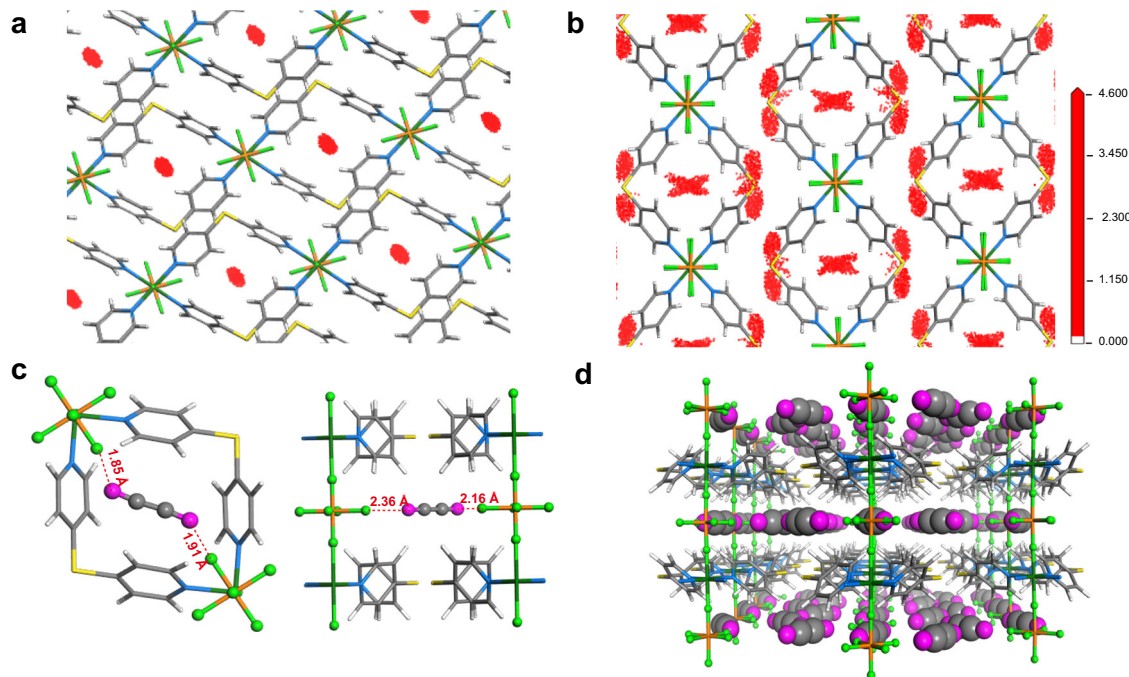

**Fig. 3 Simulated density distribution of C₂H₂ in SIFSIX-dps-Cu.** C₂H₂ distributions in SIFSIX-dps-Cu by GCMC simulation **a** at 1 kPa and **b** at 100 kPa and 298 K, viewed along the (CuSiF₆)∞ chains. **c** DFT-D calculated C₂H₂ binding mode in SIFSIX-dps-Cu. **d** Packing mode of C₂H₂-loaded SIFSIX-dps-Cu structure.

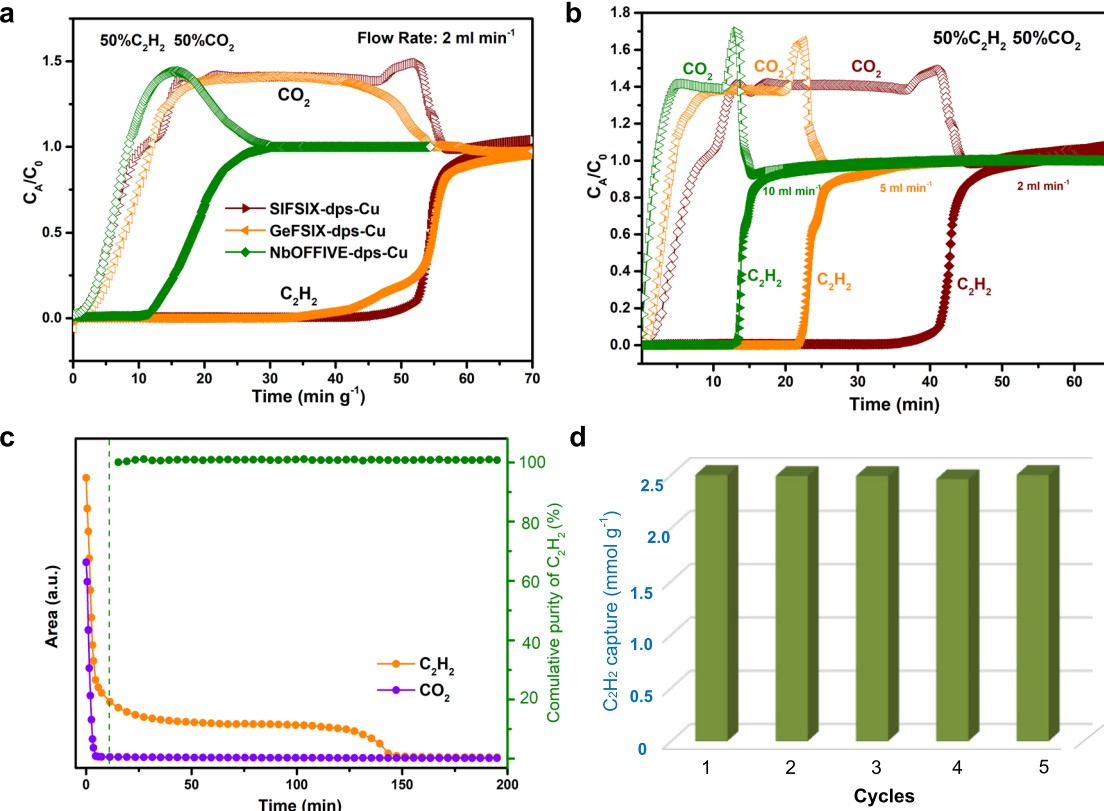

**Fig. 4 C₂H₂/CO₂ separation.** Breakthrough curves of C₂H₂/CO₂ (50/50) in **a** representative MOFs with a 2 ml min⁻¹ flow rate at 298 K. **b** The breakthrough curves with different flow rates at 298 K. **c** The signals of desorbed gases from SIFSIX-dps-Cu. **d** Cycling stability of SIFSIX-dps-Cu for C₂H₂/CO₂ (50/50) separation.

saturation with a capacity of 3.72 mmol g$^{-1}$ at 0.5 bar and 298 K, indicating that $CO_2$ cannot diffuse into the framework of SIFSIX-dps-Cu (Supplementary Fig. 39). At 5 and 10 ml min$^{-1}$, the roll-up phenomenon becomes significant that might induce by faster gas displacement at a higher flow rate (Fig. 4b). Moreover, the adsorbent could be completely regenerated with a He flows rate of 10 ml min$^{-1}$ at 298 K. In the desorption process, $CO_2$ will be eluted immediately, and then high-purity $C_2H_2$ product (≥99.9%) can be collected in the interior of the packed column for 137 min (Fig. 4c). This result also demonstrated the negligible co-adsorption of $CO_2$. For the minor inconsistency between the $CO_2$ breakthrough curve and equilibrium $CO_2$ isotherm of NbOFFIVE-dps-Cu, it can be attributed to the preferential capture of $C_2H_2$ from the dynamic gas flow by this MOF that inhibits $CO_2$ adsorption, as indicated by distinctly different adsorption heats ($C_2H_2$: 53.6 kJ mol$^{-1}$, $CO_2$: 28.8 kJ mol$^{-1}$) and dual-components adsorption result (Supplementary Fig. 17). The corresponding desorption curves of NbOFFIVE-dps-Cu in the regeneration process after $C_2H_2/CO_2$ breakthrough have also confirmed negligible $CO_2$ co-adsorption (Supplementary Fig. 40).

Furthermore, the $C_2H_2$ productivity of SIFSIX-dps-Cu calculated from the breakthrough curve is 2.48 mmol g$^{-1}$, which outperforms those of CPL-1-NH$_2$ (1.38 mmol g$^{-1}$)[41], and is comparable to SIFSIX-3-Ni (2.5 mmol g$^{-1}$)[44]. And the $C_2H_2$ productivity of GeFSIX-dps-Cu and NbOFFIVE-dps-Cu was calculated to be 2.36 and 0.83 mmol g$^{-1}$, respectively, which are significantly superior to that of UTSA-300a (0.77 mmol g$^{-1}$)[13]. In particular, these results validate the flexible-robust pore space of these dynamic MOFs for simultaneous high $C_2H_2$ capacity and $C_2H_2/CO_2$ selectivity. It should be noted that the $C_2H_2$ productivity of the three MOFs calculated from breakthrough curves is lower than those of corresponding maximum adsorption capacity (4.57, 4.04, and 1.65 mmol g$^{-1}$ for SIFSIX-dps-Cu, GeFSIX-dps-Cu, and NbOFFIVE-dps-Cu, respectively), which might be attributed to the inter-framework diffusional resistances. For practical industrial applications, the adsorbents are expected to show good recyclability. Thus, five successive $C_2H_2/CO_2$ dynamic breakthrough experiments were carried out on SIFSIX-dps-Cu, GeFSIX-dps-Cu, and NbOFFIVE-dps-Cu at three flow rates, and negligible deteriorations in breakthrough time and working capacity during five cycles indicates their outstanding recyclability (Fig. 4d and Supplementary Fig. 38). Furthermore, PXRD and sorption studies upon various conditions show that SIFSIX-dps-Cu can maintain its crystallinity in water, several organic solvents or under moisture for at least 7 days (Supplementary Fig. 41).

## Discussion

The adsorption behaviors in a series of isoreticular MOFs have been successfully controlled as a result of pore size adjustment through altering the anionic linkers, and thus realizing highly efficient $C_2H_2/CO_2$ separations. After tuning with dual functionality namely appropriate pore size and specific functional sites, novel adsorbent variants can exhibit both excellent $C_2H_2$ uptake and $C_2H_2/CO_2$ selectivity at ambient conditions. The precise pore engineering would apply to many other MOFs regarding basic principles in MOF chemistry. This work illustrates a good example to realize high-performance materials for molecular recognition and will inspire future designs on novel porous materials.

## Methods

All reagents were purchased from commercial companies and used without further purification.

**Synthesis of SIFSIX-dps-Cu**. The sample was prepared according to ref. [39], and reproduced here for completeness. A methanol solution (5.0 ml) of dps (0.054 g, 0.286 mmol) was slowly added to an aqueous solution (5.0 ml) of Cu (BF$_4$)$_2$·xH$_2$O (0.066 g, 0.26 mmol) and (NH$_4$)$_2$SiF$_6$ (0.046 g, 0.26 mmol) at room temperature, the mixture was kept undistured at room temperature for 48 h. Then the purple powder was washed with methanol and dried under a high vacuum at room temperature for 24 h.

**Synthesis of GeFSIX-dps-Cu**. A methanol solution (5.0 ml) of dps (0.054 g, 0.286 mmol) was slowly added to an aqueous solution (5.0 ml) of Cu (BF$_4$)$_2$·xH$_2$O (0.066 g, 0.26 mmol) and (NH$_4$)$_2$GeF$_6$ (0.058 g, 0.26 mmol) at room temperature, the mixture was kept undistured at room temperature for 48 h. Then the purple powder was washed with methanol and dried under a high vacuum at room temperature for 24 h.

The single crystals of GeFSIX-dps-Cu were synthesized by the slow diffusion of a methanol solution (1.0 ml) of dps (0.011 g, 0.057 mmol) into an aqueous solution (1.0 ml) of Cu(BF$_4$)$_2$·xH$_2$O (0.013 g, 0.052 mmol) and (NH$_4$)$_2$GeF$_6$ (0.012 g, 0.052 mmol) at room temperature in a watch glass without stirring. Specifically, 0.5 ml of 1:1 methanol/H$_2$O was layered between the top and bottom solutions to slow the rate of reaction. Light purple and rectangular prismatic crystals formed after 4 days.

**Synthesis of NbOFFIVE-dps-Cu**. A methanol solution (5.0 ml) of dps (0.054 g, 0.286 mmol) was slowly added to an aqueous solution (5.0 ml) of Cu (BF$_4$)$_2$·xH$_2$O (0.066 g, 0.26 mmol) and (NH$_4$)$_2$NbF$_6$ (0.059 g, 0.26 mmol) at room temperature, the mixture was kept undistured at room temperature for 48 h. Then the purple powder was washed with methanol and dried under a high vacuum at room temperature for 24 h.

The single crystals of NbOFFIVE-dps-Cu were synthesized by the slow diffusion of a methanol solution (1.0 ml) of dps (0.011 g, 0.057 mmol) into an aqueous solution (1.0 ml) of Cu(BF$_4$)$_2$·xH$_2$O (0.013 g, 0.052 mmol) and (NH$_4$)$_2$NbF$_6$ (0.012 g, 0.052 mmol) at room temperature in a watch glass without stirring. Specifically, 0.5 ml of 1:1 methanol/H$_2$O was layered between the top and bottom solutions to slow the rate of reaction. Light purple and rectangular prismatic crystals formed after 4 days.

**X-ray diffraction structure analysis**. PXRD patterns were measured by a PANalytical Empyrean Series 2 diffractometer with Cu Kα radiation with a step size of 0.0167°, a scan time of 15 s per step, and 2θ ranging from 5 to 90° at room temperature.

**Single-crystal X-ray diffraction**. Single crystal X-ray diffraction data for GeFSIX-dps-Cu and NbOFFIVE-dps-Cu were collected at 193(2) K on a Bruker-AXS D8 VENTURE diffractometer equipped with a PHOTON-100/CMOS detector (GaKα, λ = 1.3414 Å). Indexing was performed using APEX2. SaintPlus 6.01 was used to complete data integration and reduction. The multi-scan method implemented in SADABS was used to conduct absorption correction. XPREP implemented in APEX2.1 was used to determine the space group. The structures were solved by direct methods and refined by nonlinear least-squares on $F^2$ (method) with SHELXL-97 contained in APEX2, OLEX2 v1.1.5, and WinGX v1.70.01 program packages. All non-hydrogen atoms were refined anisotropically. The Squeeze routine implemented in Platon was used to treat the contribution of disordered solvent molecules as diffuse.

**The thermogravimetric analysis (TGA)**. The thermogravimetric analysis (TGA) data were collected in a NETZSCH Thermogravimetric Analyzer (STA2500) from 25 to 700 °C with a heating rate of 10 °C/min.

**Gas sorption measurements**. A Micromeritics ASAP 2460 adsorption apparatus was used to measure gas adsorption isotherms. In order to remove all the guest solvents in the framework, the fresh powder samples were evacuated under a high vacuum at room temperature (298 K) for 72 h. Liquid nitrogen and dry ice-acetone bath were used for adsorption isotherms at 77 or 196 K. The helium gas was used to determine the free space of the system. The degas procedure was repeated on the same sample between measurements for 24 h.

**Calculation of isosteric heat of adsorption ($Q_{st}$)**. The experimental adsorption enthalpy ($Q_{st}$) was applied to evaluate the binding strength between adsorbent and adsorbate, defined as

$$Q_{st} = RT^2 \left( \frac{\partial lnp}{\partial T} \right) \tag{1}$$

The isosteric heat of adsorption, $Q_{st}$ is determined using the pure component isotherm fits using the Clausius–Clapeyron equation, where $Q_{st}$ (kJ mol$^{-1}$) is the isosteric heat of adsorption, $T$ (K) is the temperature, $p$ (kPa) is the pressure, and $R$ is the gas constant.

**DFT calculations**. The first-principles DFT calculations were proceeded using the Quantum-Espresso package[52]. van der Waals interactions were illustrated by the calculation with a semiempirical addition of dispersive forces to the conventional DFT[53]. Vanderbilt-type ultrasoft pseudopotentials and generalized gradient approximation (GGA) with a Perdew–Burke–Ernzerhof were used for exchange-correlation. We found that cutoff energy of 544 eV and a $2 \times 2 \times 2$ k-point mesh (generated using the Monkhosrt–Pack scheme) were enough for the total energy to converge within 0.01 meV/atom. Fully open structures were used for the calculations of binding energy. First, the structure was optimized. Afterward, the various guest gas molecules were placed to various locations of the pore structure, followed by a full structural relaxation. To gain the gas binding energy, an isolated gas molecule that was placed in a supercell (with the same cell dimensions as the MOF crystal) was also relaxed. The static binding energy (at $T = 0$ K) was then calculated using $E_B = E(MOF) + E(gas) - E(MOF + gas)$.

**GCMC simulations**. The GCMC simulations which were carried out to investigate the adsorbed capacity of wavy layered MOFs for $C_2H_2/CO_2$ at 298 K from 0.001 to 100 kPa were performed by sorption code in MS software. Activated structures were used for the simulation of adsorption before gate-opening, whereas fully open structures were used for the simulation of adsorption after gate-opening. We used a simulation box with a $1 \times 1 \times 1$ crystallographic unit cell. During the simulations, in order to guarantee the equilibration and to sample the desired properties, $4 \times 10^6$ steps were performed. In all simulations, a rigid framework assumption was employed. We describe the interactions using the Dreiding forcefield parameter12, Lenard–Jones 12-6 potential was used to depict the van der Waals interaction with a cutoff of 15.5 $Å^{12}$.

The GCMC simulations were performed in the $NVT$ ensemble to calculate the isosteric heats of adsorption $Q_{st}$. The internal energy $\Delta U$ was computed during the simulation, which is directly related to $Q_{st}$. The isosteric heat of adsorption $Q_{st}$ was calculated from

$$Q_{st} = RT - \frac{\langle U_{ff} N \rangle - \langle U_{ff} \rangle \langle N \rangle}{\langle N^2 \rangle - \langle N \rangle \langle N \rangle} - \frac{\langle U_{sf} N \rangle - \langle U_{sf} \rangle \langle N \rangle}{\langle N^2 \rangle - \langle N \rangle \langle N \rangle} \quad (2)$$

where $R$ is the gas constant, $N$ is the number of molecules adsorbed, and $\langle \rangle$ indicates the ensemble average. The $U_{ff}$ in the first and second terms are the contributions from the molecular thermal energy and adsorbate-adsorbate interaction energy, respectively. The $U_{sf}$ in the third term is the contribution from the adsorbent–adsorbate interaction energy.

**IAST calculations**. In order to calculate the selective sorption performance for SIFSIX-dps-Cu, GeFSIX-dps-Cu, and NbOFFIVE-dps-Cu toward the separation of binary mixed gases, the fitting of single-component $C_2H_2$ and $CO_2$ adsorption isotherms were carried out based on the DSLF model. The fitting parameters of the DSLF equation are displayed in Supplementary Table 7. Adsorption isotherms and gas selectivities of mixed $C_2H_2/CO_2$ (50/50, v/v) at 298 K were predicted using the IAST. The results are shown in Supplementary Figs. 22–27.

DSLF model is listed below

$$N = N_1^{max} \times \frac{b_1 p^{1/n1}}{1 + b_1 p^{1/n1}} + N_2^{max} \times \frac{b_2 p^{1/n2}}{1 + b_2 p^{1/n2}} \quad (3)$$

Where $p$ (unit: kPa) is the pressure of the bulk gas at equilibrium with the adsorbed phase, $N$ (unit: mol/kg) is the adsorbed amount per mass of adsorbent, $N_1^{max}$ and $N_2^{max}$ (unit: mmol/g) are the saturation capacities of two different sites, $b_1$ and $b_2$ (unit: 1/kPa) are the affinity coefficients of these sites, and $n_1$ and $n_2$ represent the deviations from an ideal homogeneous surface.

The adsorption selectivity for the mixtures $C_2H_2/CO_2$ is defined by

$$S_{ads} = \frac{q_1/q_2}{p_1/p_2} \quad (4)$$

were calculated according to the IAST model proposed by Myers[54,55], in the above equation, $q_1$ and $q_2$ are the absolute component loadings of the adsorbed phase in the mixture. These component loadings are also termed uptake capacities.

**Transient breakthrough experiments**. The breakthrough experiments were carried out in a homemade apparatus. The feeding streams are gas-mixtures of 50/50 (v/v) $C_2H_2/CO_2$ with a flow rate of 2 ml min$^{-1}$ (298 K and 1.01 bar). The mass packed in the sample holder was: SIFSIX-dps-Cu (0.7881 g), GeFSIX-dps-Cu (0.7624 g), and NbOFFIVE-dps-Cu (0.7313 g). Activated samples were packed into a $\Phi 6.3 \times 140$ mm stainless steel column. A carrier gas (He ≥ 99.999%) was used to purge the adsorption bed for about 12 h at room temperature. A mass flow meter was used to regulate the gas flows, and the outlet gas from the column was monitored using mass spectrometry (Hidden, UK).) After each separation test, the sample was regenerated with a He flow of 15 ml min$^{-1}$ at 298 K. Ultrahigh purity grade helium (99.999%), acetylene (>99%), carbon dioxide (99%), and nitrogen (99.999%) were purchased from Nanchang Guoteng Gas Co., Ltd. (China).

## Data availability

All data supporting the finding of this study are available within this article and its Supplementary Information. Crystallographic data for the structures in this article have been deposited at the Cambridge Crystallographic Data Centre under deposition Nos. CCDC 2060207 (GeFSIX-dps-Cu) and 2060208 (NbOFFIVE-dps-Cu). Copies of the data can be obtained free of charge from www.ccdc.cam.ac.uk/data_request/cif. Source data that support the findings of this study are available from the corresponding author upon request. Additional graphics, model fitting, and calculations are available within its Supplementary Information.

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

## Acknowledgements

The research work was supported by the National Natural Science Foundation of China Nos. 21908090 and 22168023 (J.W.), and 22008099 (Z.Y.); Hundred Talents Program of Sun Yat-Sen University (R.-B.L.); the Natural Science Foundation of Jiangxi Province No. 20192ACB21015 (J.W.); and Welch Foundation AX-1730 (B.C.).

## Author contributions

J.W., B.C., S.D., and R.-B.L. conceived the project and designed the research, and co-wrote the paper. J.W. and Y.Z. designed the MOF materials. J.W. and Y.Z. carried out the materials synthesis and adsorption experiments. S.Y. and X.L. carried out conducted column breakthrough measurements. P.X. and S.C. performed the IAST calculation and simulation. Z.Z. and Q.D. collected and analyzed X-ray diffraction data. All authors contributed to the discussion of results and commented on the paper.

## Competing interests

The authors declare no competing interests.
