## [Peer Review File · Nature Communications]

Fine Pore Engineering in a Series of Ultramicroporous Metal-Organic Frameworks for Efficient C₂H₂/CO₂ SeparationREVIEWER COMMENTS

Reviewer #1 (Remarks to the Author):

Wang and co-authors presented a nice work on synthesis of three isorecticular MOFs, which can realize highly efficient C₂H₂/CO₂ separation. The phenomenon is very interesting, because the remarkable adsorption performances enhancement arises from the subtle structure differences. Combined experimental and theoretical tools were performed to understand the mechanism, however, there are some problems during the analyses.

1. First of all, the SIFSIX-dps-Cu is not new. The authors have reported its excellent separation properties for other gas mixture including C₂H₂/C₂H₄ and CO₂/C₂H₂/C₂H₄ in JACS, 2020, 142, 9744-9751. Thus, it is not surprising to observe the similar adsorption behaviors for C₂H₂/CO₂ using three isorecticular MOFs.
2. Although the IAST was usually used to estimate the adsorption for gas mixture, however, the theory is based on the single component adsorption isotherms. The question is that the single component adsorption isotherms at the different pressures for C₂H₂ and CO₂ gases are actually based on the different frameworks because the frameworks are flexible. Although the CO₂ cannot enter the type-II site at relatively low pressure, the pore is actually opened as a result of the C₂H₂ adsorption. Therefore, for the case of the gas mixture adsorption, the CO₂ adsorption at relatively low pressure is possible, leading to a lower selectivity. It is reasonable to conclude that the selectivity for C₂H₂/CO₂ should be overestimated using the IAST.
3. For the transient breakthrough experiments, it will be great if the adsorption loading for both CO₂ and C₂H₂ gases could be calculated at the end of the breakthrough. Then, the values could be used to compare the adsorption loading using single gas sorption isotherms. Another question is about the abrupt curves near the end of the breakthrough for CO₂ in SIFSIX-dps-Cu, which is not observed for CO₂ using GeFSIX-dps-Cu. Why?
4. Although the DSLF fitting data match well with the experimental data, some of the fitted parameters are nonsense. For example, the saturation loading at site 2 (N₂max (C₂H₂)) for NbOFFIVE-Cu-dps is 1964.49786 mmol/g, the N₂max (CO₂) for SIFSIX-Cu-dps is 194.20461 mmol/g, all of which are physically impossible for gas loading. However, the authors ignore these problems and no discussion was made.
5. The DFT calculated adsorption sites for C₂H₂ are given in the manuscript, but the DFT adsorption sites for CO₂ were not discussed. This is very important to understand the adsorption mechanism. The authors reported that the DFT-D calculations unravel the rotation of the ligands due to the C₂H₂ adsorption. What about the CO₂ adsorption? It would be necessary to discuss the CO₂ adsorption sites in the frameworks, the impact of the CO₂ adsorption on the rotation of the ligands. Especially, the gate-open effect for CO₂ adsorption in SIFSIX-dps-Cu suggests that there is a structural change with and without the CO₂ adsorption.
6. The sorption code was used to simulate the gas adsorption for the MOFs with rigid framework, which is clearly not able to simulate the flexible properties of the frameworks. Therefore, the adsorption behavior with pressure from 0.001 to 100 kPa is clearly underestimated by the GCMC simulations, which is a serious problem for modeling the adsorption sites for both C₂H₂ and CO₂. However, the authors did not comment on these issues.

Reviewer #2 (Remarks to the Author):

In this manuscript (NCOMMS-21-19680), Banglin Chen and coworkers demonstrate the C₂H₂/CO₂ separation performance of a series of SIFSIX-dps-Zn variant structures by altering anionic linkers and metal nodes. The slight differences in the size of intralayer cavity and interlayer cavity lead to the great improvement in C₂H₂ adsorption performance and C₂H₂/CO₂ selectivity. The different adsorption behavior of C₂H₂ is shown in adsorption isotherms and explained through the analysis of structural changes. Modeling simulation studies nicely explain the observations. This is an interesting piece of work, and it could be considered for publication in Nature Communications, if specific

comments below can be well addressed.

1. "In industry, C₂H₂ is produced by partial CH₄ combustion and thermal hydrocarbon cracking, in which the main impurity carbon dioxide (CO₂) would show great impact upon the subsequent industrial processes." Actually, CO₂ is not the main impurity, only a very small amount of CO₂ exists in the production in industry.
2. The description about the XRD pattern changes and the corresponding structure changes of as-synthesized, activated and C₂H₂-loaded samples should be provided in the manuscript.
3. The BET surface area plots for the samples should be presented.
4. The authors show the differences of the inclined angle between anion pillar and dps ligand before and after C₂H₂ loading on samples. More experiment details about measuring conditions and Rietveld refinements should be provided. It seems that the C₂H₂ binding sites were located in the cavity I by Rietveld refinement in Figure S14, could the authors provide further information about C₂H₂ molecules in the cavity?
5. The C₂H₂/CO₂ selectivity on UTSA-300 should be mentioned in the manuscript and provided in the Figure 2(f).
6. The authors use a dual site Langmuir-Freundlich model to fit the C₂H₂ and CO₂ isotherm data, what is the physical meaning of the parameters? Some parameters in Table S3 are too large, especially the value of N.
7. It should be clarified that the structure used in the modeling simulation is activated structure or before activated.
8. In the density distribution of C₂H₂ at 100 kPa, the density of C₂H₂ neighboring S atom is high, the interaction between S and C₂H₂ should be considered.
9. "Taking SIFSIX-dps-Cu as the representative sample, in Site I, the C₂H₂ molecule is bound by two F atoms from two different SiF₆²⁻ units with H...F (1.85 Å) and C-H...F (1.91 Å) bonds (Figure 3c)." "At Site II, the C₂H₂ molecule locates in the interlayer cavities and is asymmetrically trapped with the H...F (2.16 Å) and C-H...F (2.36 Å) host-guest interactions (Figure 3c)." The description of the interaction between C₂H₂ and SiF₆²⁻ should be same in the manuscript, why two term H...F and C-H...F were used? .
10. In Site I, the distance (C-H...F) between C₂H₂ and SiF₆²⁻ is 1.85 Å and 1.91 Å, respectively, and the corresponding adsorption energy is calculated to be 60.3 kJ mol⁻¹. In Site II, the distance (C-H...F) between C₂H₂ and SiF₆²⁻ is 2.16 Å and 2.36 Å, and the corresponding adsorption energy is calculated to be 67.5 kJ mol⁻¹. Please explain why the distance (C-H...F) between C₂H₂ and SiF₆²⁻ in Site I is shorter than that in Site II, the corresponding adsorption energy in Site II is stronger than that in Site I.
11. The Clausius-Clapeyron plot used to calculate the Q_{st} values should be presented.
12. There is a "roll-up" of CO₂ in breakthrough curves on SIFSIX-dps-Cu, and it becomes more obvious with the increasing of flow rate, please explain it.
13. The C₂H₂ isotherms at 298 K of NbOFFIVE-dps-Cu in this work and ZUJ-220 (Shen, J., He, X., Ke, T. et al. Nat Commun 11, 6259 (2020)) is very different. Is it possible that the water molecules is still in the framework? As the NbOFFIVE-dps-Cu sample is just washed by the methanol and evacuated at room temperature.
14. In the synthesis of NbOFFIVE-dps-Cu, (NH₄)₂NbF₆ was used. Does (NH₄)₂NbF₆ react with Cu(BF₄)₂·xH₂O to produce CuNbOF₅? Please check it.
15. "Binary mixture (C₂H₂/CO₂, 50/50, v/v) were injected into a packed column with a flow rate of 2.0 mL min⁻¹ and the clean separations of C₂H₂/CO₂ mixtures were realized by all dynamic layered materials (Figure 27)." The Figure 27 should be Figure S28.

Reviewer #3 (Remarks to the Author):

In this manuscript, the authors report the enhancement of adsorption capacity and selectivity of acetylene in SIFSIX-dps-M MOFs by changing the metal (M). Among three isoreticular compounds investigated, SIFSIX-dps-Cu (a structure previously reported by the same authors in J. Am. Chem. Soc. 2020, 142, 9744) has reached an uptake of 4.57 mmol/g for C₂H₂ at 298 K and 1 bar and the

highest IAST selectivity of 1787 for C₂H₂/CO₂. The authors also analyzed host-guest interactions using dispersion-corrected density functional theory and grand canonical Monte Carlo simulations. The following questions should be addressed.

1. There is too little data on stability test. Putting the sample in air for 24 h is not enough because it did not offer the exact humidity which is very important to the stability of material. The test should be done for a longer time (e.g. one week) under high humidity to prove the high stability. I suggest to add PXRD results after putting samples under high humidity, being soaked in water at different temperature and PH, and being soaked in other common solvents.
2. On page 5, the authors state that "... the activated crystal structures are determined by Rietveld refinements (Figures 1 and S7). Figure S7 shows PXRD patterns but the refinement details and refined structure data are not provided. These should be added in the supporting information.
3. UTSA-300-Zn possesses a very high C₂H₂/CO₂ IAST selectivity of 743 (J. Am. Chem. Soc. 2017, 139, 8022, higher than other materials in Figure 2e and 2f except SIFSIX-dps-Cu. However, the author did not refer this material in the Figure 2e and Figure 2f. I think the author should add UTSA-300-Zn into these two figures. Also, the conditions (e.g. temp and pressure) in Figure 2e and 2f should be specified.
4. The acetylene binding modes analysis is not new. It was already reported in the previous work (J. Am. Chem. Soc. 2020, 142, 9744). However, the DFT-D calculated binding modes (Site I and Site II) show very different F...H distances in the current paper (Figure 3c) and previous JACS paper (Figure 4d). Please explain in the revised paper.
5. The CO₂ breakthrough curves on SIFSIX-dps-Cu (Figure 4a) is much complicated than that of GeFSIX-dps-Cu and NbOFFIVE-dps-Cu. Can you explain the two steps before 15 min/g and the further roll up at 45 min/g in the breakthrough curves of CO₂ on SIFSIX-dps-Cu?
6. The uptake amount of C₂H₂ can be obtained from the simulation studies. This value should be compared with the experimental C₂H₂ uptake and discussed in the paper. Also, the sentence "Figure 4d shows the optimized structure of C₂H₂-loaded SIXSIX-dps-Cu sample ..." on page 9 has an error. "Figure 4d" should be "Figure 3d".
7. Compare Figure S7 with Figure S15, the PXRD peaks changed after loading with C₂H₂, revealing gate-opening of the pore. A more detailed explanation should be given to relate the changes in the peak position and in the structure, namely how the structure is changed?
8. In Figure S9, the BET surface areas are estimated based on CO₂ sorption data. I wonder if the result for NbOFFIVE-dps-Cu is accurate enough. How are the points chosen from the isotherm for fitting with such a shape? What is the pore size distribution? Is it consistent with the pore size value shown in Figure 1?
9. Page 13. The authors state that "Single crystal X-ray diffraction data for SIFSIX-dps-Cu, GeFSIX-dps-Cu, and NbOFFIVE-dps-Cu were collected at 193(2) K ...". Since the single crystal data of SIFSIX-dps-Cu has already been reported by the authors before, why is there a need to correct data again? If the data should in Table 2 are from the JACS 2020 paper, then it should be so indicated in the table, and SIFSIX-dps-Cu should be removed from the above statement.
10. None of Supporting Tables are mentioned in the main text. They need to be cited.
11. What does "transit breakthrough experiments" mean?

Author Responses to Reviewers' Comments

Reviewer #1 (Remarks to the Author):

Overall Comment. Wang and co-authors presented a nice work on synthesis of three isoreticular MOFs, which can realize highly efficient C_2H_2/CO_2 separation. The phenomenon is very interesting, because the remarkable adsorption performances enhancement arises from the subtle structure differences. Combined experimental and theoretical tools were performed to understand the mechanism, however, there are some problems during the analyses.

Author response: Thank you very much for your very important and constructive comments, which have significantly helped us to improve the quality of this work.

Comment 1. First of all, the SIFSIX-dps-Cu is not new. The authors have reported its excellent separation properties for other gas mixture including C_2H_2/C_2H_4 and $CO_2/C_2H_2/C_2H_4$ in JACS, 2020, 142, 9744-9751. Thus, it is not surprising to observe the similar adsorption behaviors for C_2H_2/CO_2 using three isoreticular MOFs.

Author response: Thank you very much for your very valuable and constructive comments. After we thoroughly and carefully read your comments, we realized that we did not state the research background clearly. Accordingly, we thus have quite significantly revised the manuscript to make suitable claims. It is true that SIFSIX-dps-Cu is not new and reported with C_2H_2 -exclusive adsorption from $C_2H_2/C_2H_4/CO_2$ mixture (Reference 39), though it was a preliminary result about the influence of CO_2 on C_2H_2/C_2H_4 separation. The pore structure in the series of isoreticular MOFs are varying in sub-angstrom scale (≤ 0.9 Å, Figure 1) decorating with the same fluorite sites, so it is true to speculate similar adsorption behaviors for C_2H_2/CO_2 using these isoreticular MOFs; however, a systematical study on fine-tuning of pore structure enables isoreticular MOFs to simultaneously achieve high capture capacity and adsorption selectivity. Both the separation performance and adsorption mechanism have been thoroughly and carefully studied to address the more challenging C_2H_2/CO_2 separation. We have thus modified the manuscript title and revised the introduction

section clearly. I would like to thank you once again for your very important and critical comments, which have significantly helped us to improve the quality of this work.

Modifications:

Manuscript: Title

“Fine Pore Engineering in a Series of **Isorecticular** Metal-Organic Frameworks for Highly Efficient C₂H₂/CO₂ Separation”

Manuscript: Introduction section, Paragraph 2

“...which has been demonstrated by [Cu(dps)₂(SiF₆)] (SIFSIX-dps-Cu) for size-exclusive adsorption of C₂H₂ from C₂H₄.³⁹ To achieve simultaneously high capacity and separation selectivity for more challenging C₂H₂/CO₂ separation, a systematical study on fine-tuning of pore structure...”

Comment 2. Although the IAST was usually used to estimate the adsorption for gas mixture, however, the theory is based on the single component adsorption isotherms. The question is that the single component adsorption isotherms at the different pressures for C₂H₂ and CO₂ gases are actually based on the different frameworks because the frameworks are flexible. Although the CO₂ cannot enter the type-II site at relatively low pressure, the pore is actually opened as a result of the C₂H₂ adsorption. Therefore, for the case of the gas mixture adsorption, the CO₂ adsorption at relatively low pressure is possible, leading to a lower selectivity. It is reasonable to conclude that the selectivity for C₂H₂/CO₂ should be overestimated using the IAST.

Author Response: Thank you very much for the constructive comments. It is true that there are arguments on co-adsorption of impurity gas in flexible MOFs for separating a gas mixture. And the IAST calculation could overestimate the separation selectivity, which could be used for qualitative comparison only. Therefore, we have revised our statement on the IAST selectivity.

As for the possible CO₂ co-adsorption, this is indeed a very important and interesting point. In fact, there are studies of flexible MOFs for gas separation that did

find the co-adsorption of mixed gases after pore opening occurred (J. Am. Chem. Soc. 2009, 131, 17490). It is noted that there is a small roll-up of the breakthrough curve for CO₂ in SiFSIX-dps-Cu, being consistent with the minor CO₂ uptake from single-component sorption isotherms. That indicates a minor co-adsorption of CO₂ during the dynamic capture process, and finally desorbed by C₂H₂ when the latter starting breakthrough out (J. Am. Chem. Soc. 2009, 131, 17490). On the other hand, for SIFSIX-dps-Cu after breakthrough experiments, the desorption concentration curves (Figure 4c) of both gases indicate that there was no noticeable CO₂ co-adsorption in the final step whereas only C₂H₂ with a purity of $\geq 99.9\%$. Furthermore, mixed-components sorption isotherms of the three isostructural MOFs for C₂H₂/CO₂ (50/50, mol/mol) have also been collected (Figure S17). Compared with the single component sorption results, their adsorption capacity and threshold pressure are basically not changed. All the above results indicate the preferential adsorption of C₂H₂ after the breakthrough for C₂H₂/CO₂ mixture. For clarity, these results and discussions have been added to the revised manuscript.

Modifications:

Manuscript: Page 7

“To investigate the adsorption phenomena for gas mixture, mixed-components adsorption isotherms of the three isorecticular MOFs for C₂H₂/CO₂ (50/50, mol/mol) have also been collected (Figure S17). Compared with the single component sorption results, their adsorption capacity and threshold pressure are basically not changed, which confirms the preferential adsorption of C₂H₂ from C₂H₂/CO₂ mixture.⁴³”

Manuscript: Page 8

“Prompted by the dramatic uptake differences, the ideal adsorbed solution theory (IAST) was applied to qualitatively estimate the C₂H₂/CO₂ selectivity, while the adsorption isotherms are fitted by the dual-site Langmuir-Freundlich (DLSF) equation with excellent accuracy (Figures S18-29, Table S7).”

Manuscript: Page 10

“For small roll-up of the breakthrough curves for CO₂ in SiFSIX-dps-Cu, it can be attributed to the desorption of CO₂ induced by C₂H₂, which indicates a minor co-adsorption of CO₂ during the dynamic capture process, and finally displaced by C₂H₂. That phenomenon is consistent with the minor CO₂ uptake from single-component sorption isotherms.”

Supplementary information:

Figure S17. Sorption isotherms of SIFSIX-dps-Cu, GeFSIX-dps-Cu, and NbOFFIVE-dps-Cu for pure C₂H₂ (solid symbol) and equimolar mixture of C₂H₂/CO₂ (50/50, mol/mol, empty symbol) at 298 K, respectively.

Comment 3. For the transient breakthrough experiments, it will be great if the adsorption loading for both CO₂ and C₂H₂ gases could be calculated at the end of the breakthrough. Then, the values could be used to compare the adsorption loading using single gas sorption isotherms. Another question is about the abrupt curves near the end of the breakthrough for CO₂ in SIFSIX-dps-Cu, which is not observed for CO₂ using GeFSIX-dps-Cu. Why?

Author Response: Thank you for pointing this out. The C₂H₂ productivity calculated from the breakthrough curves are 2.48, 2.36, and 0.83 mmol g⁻¹ for SIFSIX-dps-Cu, GeFSIX-dps-Cu, and NbOFFIVE-dps-Cu, respectively. Accordingly, we have clarified these results and compared with the corresponding maximum adsorption capacity in the

revised manuscript.

For the small roll-up of the breakthrough curve for CO₂ in SIFSIX-dps-Cu, it can be attributed to the desorption of CO₂ induced by C₂H₂, when the concentration front of C₂H₂ advances through the column. That phenomenon is consistent with the minor CO₂ uptake from single-component sorption isotherms, which indicates a minor co-adsorption of CO₂ during the dynamic capture process, and finally desorbed by C₂H₂ when the latter starting a breakthrough out (J. Am. Chem. Soc. 2009, 131, 17490). That is why there is no co-adsorption of CO₂ observed during the regeneration step. In contrast, there was no co-adsorption of CO₂ in GeFSIX-dps-Cu during either single-component gas sorption or transient breakthrough adsorption process, which leads to a relatively smooth concentration curve. We have added these discussions to the revised main text.

Modifications:

Manuscript: Page 10-11

“Furthermore, the C₂H₂ productivity of SIFSIX-dps-Cu calculated from the breakthrough curve is 2.48 mmol g⁻¹, which outperforms those of CPL-1-NH₂ (1.38 mmol g⁻¹),⁴¹ and is comparable to SIFSIX-3-Ni (2.5 mmol g⁻¹).⁴⁴ And the C₂H₂ productivity of GeFSIX-dps-Cu and NbOFFIVE-dps-Cu was calculated to be 2.36 and 0.83 mmol g⁻¹, respectively...”

“It should be noted that the C₂H₂ productivity of the three MOFs calculated from breakthrough curves are lower than those of corresponding maximum adsorption capacity (4.57, 4.04, and 1.65 mmol g⁻¹ for SIFSIX-dps-Cu, GeFSIX-dps-Cu, and NbOFFIVE-dps-Cu, respectively), which might be attributed to the inter-framework diffusional resistances.”

“For small roll-up of the breakthrough curves for CO₂ in SiFSIX-dps-Cu, it can be attributed to the desorption of CO₂ induced by C₂H₂, which indicates a minor co-adsorption of CO₂ during the dynamic capture process, and finally displaced by C₂H₂. That phenomenon is consistent with the minor CO₂ uptake from single-component

sorption isotherms.”

Comment 4. Although the DSLF fitting data match well with the experimental data, some of the fitted parameters are nonsense. For example, the saturation loading at site 2 ($N_{2max}(C_2H_2)$) for NbOFFIVE-Cu-dps is 1964.49786 mmol/g, the $N_{2max}(CO_2)$ for SIFSIX-Cu-dps is 194.20461 mmol/g, all of which are physically impossible for gas loading. However, the authors ignore these problems and no discussion was made.

Author Response: Thank you for your reminder. Accordingly, to make sure the fitting results are reasonable, we have carefully checked and re-performed corresponding DSLF fittings. The fitting results and corresponding parameters have been revised in the revised supplementary information.

Modifications:

Supplementary information:

Figure S21. DSLF fitting of the CO_2 sorption data at 298 K and 1 bar on SIFSIX-dps-Cu.

Figure S27. DSLF fitting of the C₂H₂ sorption data at 298 K and 1 bar on NbOFFIVE-dps-Cu.

Table S7. The DSLF fitting parameters for C₂H₂ and CO₂ in three isorecticular MOFs.

Adsorbents	Adsorbates	N_1^{\max} (mmol g ⁻¹)	b_1 (bar ⁻¹)	$1/n_1$	N_2^{\max} (mmol g ⁻¹)	b_2 (bar ⁻¹)	$1/n_2$
SIFSIX-Cu-dps	C ₂ H ₂ (273 K)	2.61	4.07E-12	10.8	3.56	1.86	0.342
SIFSIX-Cu-dps	C ₂ H ₂ (298 K)	3.34	1.18	0.367	2.73	1.08E-14	17.9
SIFSIX-Cu-dps	C ₂ H ₂ (313 K)	2.25	48.9	2.72	1.03	7.81	0.662
SIFSIX-Cu-dps	CO₂ (298 K)	0.985	0.652	1.41	0.313	2.15	0.575
GeFSIX-Cu-dps	C ₂ H ₂ (273 K)	3.26	3.40	0.488	1.97	7.93E-4	4.49
GeFSIX-Cu-dps	C ₂ H ₂ (298 K)	3.56	0.779	0.299	2.48	5.44E-4	5.48
GeFSIX-Cu-dps	C ₂ H ₂ (323 K)	0.675	2.69E-3	1.43	3.28	5.19	4.40
GeFSIX-Cu-dps	CO ₂ (298 K)	0.0831	1.99	7.98	5.81	0.0726	0.751
NbOFFIVE-Cu-dps	C ₂ H ₂ (273 K)	1.84	12.9	0.522	1.33	13.7	2.97
NbOFFIVE-Cu-dps	C₂H₂ (298 K)	0.805	3.47E-2	1.02	7.26	0.126	1.00
NbOFFIVE-Cu-dps	C ₂ H ₂ (323 K)	3.17	0.0744	0.328	0.531	1.09	4.96
NbOFFIVE-Cu-dps	CO ₂ (298 K)	0.108	1.62	6.90	1.07	20.2	1.17

Comment 5. The DFT calculated adsorption sites for C₂H₂ are given in the manuscript, but the DFT adsorption sites for CO₂ were not discussed. This is very important to understand the adsorption mechanism. The authors reported that the DFT-D calculations unravel the rotation of the ligands due to the C₂H₂ adsorption. What about

the CO₂ adsorption? It would be necessary to discuss the CO₂ adsorption sites in the frameworks, the impact of the CO₂ adsorption on the rotation of the ligands. Especially, the gate-open effect for CO₂ adsorption in SIFSIX-dps-Cu suggests that there is a structural change with and without the CO₂ adsorption.

Author Response: Thank you for your reminder. Accordingly, we have conducted the DFT-D calculations to study potential CO₂ adsorption sites in these frameworks as well as the impact of CO₂-loading on the structural deformation of the organic ligands. In SIFSIX-dps-Cu, CO₂ molecules locate at similar positions as those for C₂H₂ (Figure S35), interacting with pore surface through weak intermolecular interactions like electrostatic interactions (F^{δ-}...C^{δ+} 2.91–3.63 Å). According to the simulated structural configuration of CO₂-loading, the change in torsion angle between anion pillar and dps ligand caused by CO₂ loading is about 18°, which is larger than that for C₂H₂ loading (7°), in line with higher gate-opening pressure for CO₂ sorption. The results and discussion have been added to the revised manuscript and supporting information.

Modifications:

Manuscript: Page 9

“In contrast, CO₂ in SIFSIX-dps-Cu interacts with pore surface through weak intermolecular interactions like electrostatic interactions (F^{δ-}...C^{δ+} 2.91–3.63 Å), showing binding energy of 41.9 and 33.7 kJ mol⁻¹ at site I and site II, respectively (Figure S35). The change in torsion angle between anion pillar and dps ligand caused by CO₂ loading is about 18°, which is larger than that for C₂H₂ loading (7°), in line with higher gate-opening pressure for CO₂ sorption.”

Supplementary information:

Figure S35. The DFT-D calculated CO₂ binding mode in SIFSIX-dps-Cu.

Comment 6. The sorption code was used to simulate the gas adsorption for the MOFs with rigid framework, which is clearly not able to simulate the flexible properties of the frameworks. Therefore, the adsorption behavior with pressure from 0.001 to 100 kPa is clearly underestimated by the GCMC simulations, which is a serious problem for modeling the adsorption sites for both C₂H₂ and CO₂. However, the authors did not comment on these issues.

Author Response: Thank you for pointing this out. We agree that the GCMC simulations might underestimate the adsorption behavior of flexible MOFs. We realized that we did not state this clearly, and to a certain extent, might even overstate the claims. It is difficult to get the structures of intermediate states during the dynamic adsorption. But fortunately, the structures before and after C₂H₂ loading are clearly known, as illustrated by the zinc analogue UTSA-300 (Ref. 13). We thus used the open structure to investigate the potential adsorption sites in these frameworks. The adsorption distribution at different pressures (1 and 100 kPa, Figures 3 and S31-S33) indeed support our claim on the adsorption sites for C₂H₂, that also in line with those in UTSA-300 confirm by the neutron diffraction data. For clarity, we have these discussions in the revised manuscript.

Modifications:

Manuscript: Page 8-9

“To investigate the potential C₂H₂ adsorption sites in these layered MOFs, dispersion-corrected density functional theory (DFT-D) and grand canonical Monte Carlo (GCMC)

simulations are further carried out.”

“Given that it is difficult to get the structures of intermediate states during the dynamic adsorption whereas the structural change during C₂H₂ loading is similar to UTSA-300, the activated or open frameworks were thus used for simulations.”

“The distribution density of C₂H₂ was investigated firstly at 1 kPa, as shown in Figures 3a-b and S31-33, only C₂H₂ can be adsorbed in intralayer cavities (Site I) on all for isorecticular MOFs.”

“Such adsorption behavior is in line with that in UTSA-300 confirm by neutron diffraction data¹³.”

Reviewer #2 (Remarks to the Author):

Overall Comment. In this manuscript (NCOMMS-21-19680), Banglin Chen and coworkers demonstrate the C₂H₂/CO₂ separation performance of a series of SIFSIX-dps-Zn variant structures by altering anionic linkers and metal nodes. The slight differences in the size of intralayer cavity and interlayer cavity lead to the great improvement in C₂H₂ adsorption performance and C₂H₂/CO₂ selectivity. The different adsorption behavior of C₂H₂ is shown in adsorption isotherms and explained through the analysis of structural changes. Modeling simulation studies nicely explain the observations. This is an interesting piece of work, and it could be considered for publication in Nature Communications, if specific comments below can be well addressed.

Author Response: Thank you very much for your very positive and constructive comments on our manuscript.

Comment 1. “In industry, C₂H₂ is produced by partial CH₄ combustion and thermal hydrocarbon cracking, in which the main impurity carbon dioxide (CO₂) would show great impact upon the subsequent industrial processes.” Actually, CO₂ is not the main impurity, only a very small amount of CO₂ exists in the production in industry.

Author Response: Thanks for your reminder. Accordingly, we have revised the statement in the revised manuscript.

Modifications:

Manuscript: Page 2

“In industry, C₂H₂ is produced by partial CH₄ combustion and thermal hydrocarbon cracking, **in which carbon dioxide (CO₂) is a worth-noting impurity that can show** great impact upon the subsequent industrial processes^{4,5}.”

Comment 2. The description about the XRD pattern changes and the corresponding structure changes of as-synthesized, activated and C₂H₂-loaded samples should be provided in the manuscript.

Author Response: Thank you for your suggestion. The changes of PXRD patterns and corresponding structural transformation in the three isorecticular MOFs upon desolvation or C₂H₂-loading are the same as the prototypical zinc analogue UTSA-300 (Ref. 13). For the PXRD patterns of the three MOFs here, the dramatic structural transformation results in different diffraction peaks with significantly different indices, showing major changes in the original peaks at around 10°, 16–17° and 21–23°, which might be assigned to the sliding and shrinking of 2D coordination layers. Upon C₂H₂-loading, the MOFs underwent structural expansion reversing to the structure of as-synthesized status. These structural transformation processes were accompanied by ligand rotations and pore-opening (Figure S15). Accordingly, these discussions have been added to the revised manuscript and SI.

Modifications:

Manuscript: Page 6

“The changes of PXRD patterns and corresponding structural transformation in the three isorecticular MOFs upon desolvation or C₂H₂-loading are the same as the prototypical zinc analogue UTSA-300 (Figures S2-S4 and Tables S2-S5)¹³.”

Supplementary information: Figure S2

Figure S2. (a) The powder X-ray diffraction patterns of SIFSIX-dps-Cu, and (b) structural transformation between as-synthesized phase to desolvated one. For SIFSIX-dps-Cu, after activation, the original peaks at 10°, 16.6°, and 22.5° showed significant changes that might be assigned to the sliding and shrinking of 2D coordination layers. The space group changes from *Ibam* to *P2/n* during the phase transformation. a dense

structure with dispersed 0D cavities was thus obtained. Upon C_2H_2 -loading, PXRD patterns and structure configurations reverse to corresponding open state.

Figure S3. (a) The powder X-ray diffraction patterns of GeFSIX-dps-Cu, and (b) structural transformation between as-synthesized phase to desolvated one. The changes of PXRD patterns (significantly at 10° , 16° , and 21.5°) and corresponding structural transformation in GeFSIX-dps-Cu are similar to those of SIFSIX-dps-Cu.

Figure S4. (a) The powder X-ray diffraction patterns of NbOFFIVE-dps-Cu, and (b) structural transformation between as-synthesized phase to desolvated one. The changes of PXRD patterns (significantly at 10° , 16.5° , and 18°) and corresponding structural transformation in NbOFFIVE-dps-Cu are similar to those of SIFSIX-dps-Cu.

Comment 3. The BET surface area plots for the samples should be presented.

Author Response: Thank you for your reminder. Accordingly, we have added the BET surface area plots (Figure S10) in the revised supporting information.

Modifications:

Supplementary information:

Figure S10. Plots for calculation of BET surface area, based on CO₂ adsorption isotherms of (a) SIFSIX-dps-Cu, (b) GeFSIX-dps-Cu, and (c) NbOFFIVE-dps-Cu at 195 K.

Comment 4. The authors show the differences of the inclined angle between anion pillar and dps ligand before and after C₂H₂ loading on samples. More experiment details about measuring conditions and Rietveld refinements should be provided. It seems that the C₂H₂ binding sites were located in the cavity I by Rietveld refinement in Figure S14, could the authors provide further information about C₂H₂ molecules in the cavity?

Author Response: Thanks for your suggestion. The angle was measured as the torsion angle from C–N bond on pyridinyl ring to Cu–F/O bond. Accordingly, the measuring detail has been added to the relevant section in Figure S15. The details of Rietveld refinements have also been added to Figure S7 in the revised SI. As for the C₂H₂ molecules at the site I of the pore cavities, the binding configurations are illustrated in Figure S16, showing almost identical binding mode with that of UTSA-300 (Ref. 13), as well as the simulated one in Figure 3c. More detailed information can also be found in the modeling structures of C₂H₂-loaded sample (Table S2-S5). These results have been added to the revised SI.

Modifications:

Supplementary information: Figure S7

“...Rietveld structural refinement was performed on the **PXRD** data using the **GSAS** package. Due to the large number of atoms in the crystal unit cell, the ligand molecule and the gas molecule were both treated as rigid bodies during the Rietveld refinement, with the molecule orientation and center of mass freely refined. Final refinement on

the positions/orientations of the rigid bodies, thermal factors, occupancies, lattice parameters, background, and profiles converge with satisfactory R factors.”

Supplementary information: Figure S15

“...The torsion angle was measured from C–N bond on pyridinyl ring to Cu–F/O bond.”

Supplementary information: Figure S16

Figure S16. Powder X-ray diffraction patterns for Rietveld refinement of C₂H₂-loaded (a) SIFSIX-dps-Cu, (b) GeFSIX-dps-Cu, and (c) NbOFFIVE-dps-Cu. The C₂H₂ accumulation patterns at Site I and Site II of C₂H₂-loaded (d,g) SIFSIX-dps-Cu, (e,h) GeFSIX-dps-Cu, and (f,i) NbOFFIVE-dps-Cu.

Table S2. Modelling study and refinement parameters of C₂H₂-loaded MOFs.

Table S3. Coordinates of the non-hydrogen atoms in C₂H₂-loaded SIFSIX-dps-Cu.

Table S4. Coordinates of the non-hydrogen atoms in C₂H₂-loaded GeFSIX-dps-Cu.

Table S5. Coordinates of the non-hydrogen atoms in C₂H₂-loaded NbOFFIVE-dps-Cu.

Comment 5. The C_2H_2/CO_2 selectivity on UTSA-300 should be mentioned in the manuscript and provided in the Figure 2(f).

Author Response: Thanks for your reminder. Accordingly, the C_2H_2/CO_2 selectivity of UTSA-300 has been stated in the revised manuscript.

Modifications:

Manuscript: Page 8

“the calculated equimolar C_2H_2/CO_2 selectivity of NbOFFIVE-dps-Cu, GeFSIX-dps-Cu and SIFSIX-dps-Cu at 298 K and 1.0 bar are 9, 172 and 1787, respectively. In particular, the selectivity of SIFSIX-dps-Cu is higher than that of UTSA-300a (743), and much higher than other benchmark MOFs (Table S6)...”

Manuscript: Figure 2

Figure 2f. comparison about C_2H_2/CO_2 selectivity and C_2H_2 capacity of representative MOFs at 298 K and 1 bar.

Comment 6. The authors use a dual site Langmuir-Freundlich model to fit the C_2H_2 and CO_2 isotherm data, what is the physical meaning of the parameters? Some parameters in Table S3 are too large, especially the value of N .

Author Response: Thank you for your reminder. The N^{\max} values represent the theoretical adsorption capacity at specific sites, which are illustrated together with other parameters in the method details of IAST calculations. Accordingly, to make sure the fitting results are reasonable, we have carefully checked and re-performed

corresponding DSLF fittings. The fitting results and corresponding parameters have been revised in the revised supplementary information.

Modifications:

Supplementary information:

Figure S21. DSLF fitting of the CO₂ sorption data at 298 K and 1 bar on SIFSIX-dps-Cu.

Figure S27. DSLF fitting of the C₂H₂ sorption data at 298 K and 1 bar on NbOFFIVE-dps-Cu.

Table S7. The DSLF fitting parameters for C₂H₂ and CO₂ in three isorecticular MOFs.

Comment 7. It should be clarified that the structure used in the modeling simulation is activated structure or before activated.

Author Response: Thanks for the kind reminder. We have clarified the modeling in the computational simulations is the expended structure. We have highlighted this point on Page 9.

Modifications:

Manuscript: Page 9

“...whereas the structural change during C₂H₂ loading is similar to UTSA-300, the activated or open frameworks were thus used for simulations.”

Manuscript: Page 15

“Fully open structures were used for the calculations of binding energy.”

“Activated structures were used for the simulation on adsorption before gate-opening, whereas fully open structures were used for the simulation of adsorption after gate-opening.”

Comment 8. In the density distribution of C₂H₂ at 100 kPa, the density of C₂H₂ neighboring S atom is high, the interaction between S and C₂H₂ should be considered.

Author Response: Thanks for your comment. Apparently, we did not illustrate this clearly in our previous manuscript. We realized the projection of the structure in the previous figure viewed along the (CuSiF₆)_∞ chains misled your understanding. The distance between C₂H₂ molecules at site II and S atoms is about 3.94 Å, which is much longer than the sum of van der Waals radii of C and S (3.0 Å) as well as the distance between C₂H₂ molecules and SiF₆²⁻ pillars (2.36 Å). For clarity, we have revised the corresponding figure caption (Figure 3b) more clearly. And a side view of C₂H₂-loaded SIFSIX-dps-Cu with density distribution of C₂H₂ has been also added to Figure S31 in the revised supporting information.

Modifications:

Manuscript: Figure 3

“...viewed along the (CuSiF₆)_∞ chains”

Supplementary information:

Figure S31c. The side view of C₂H₂-loaded SIFSIX-dps-Cu with density distribution of C₂H₂.

Comment 9. “Taking SIFSIX-dps-Cu as the representative sample, in Site I, the C₂H₂ molecule is bound by two F atoms from two different SiF₆²⁻ units with H•••F (1.85 Å) and C–H•••F (1.91 Å) bonds (Figure 3c).” “At Site II, the C₂H₂ molecule locates in the interlayer cavities and is asymmetrically trapped with the H•••F (2.16 Å) and C–H•••F (2.36 Å) host-guest interactions (Figure 3c).” The description of the interaction between C₂H₂ and SiF₆²⁻ should be same in the manuscript, why two term H•••F and C–H•••F were used?

Author Response: Thanks for your suggestion. Accordingly, only H•••F and its corresponding distance were used to illustrate the interactions between C₂H₂ and SiF₆²⁻ in the revised manuscript.

Modifications:

Manuscript: Page 9

“...H•••F hydrogen-bond interactions (1.85–2.36 Å...)”

Comment 10. In Site I, the distance (C–H•••F) between C₂H₂ and SiF₆²⁻ is 1.85 Å and 1.91 Å, respectively, and the corresponding adsorption energy is calculated to be 60.3 kJ mol⁻¹. In Site II, the distance (C–H•••F) between C₂H₂ and SiF₆²⁻ is 2.16 Å and 2.36 Å, and the corresponding adsorption energy is calculated to be 67.5 kJ mol⁻¹. Please explain why the distance (C–H•••F) between C₂H₂ and SiF₆²⁻ in Site I is shorter than that in Site II, the corresponding adsorption energy in Site II is stronger than that in Site I.

Author Response: Thank you for clarifying and pointing this out. Apparently, we didn't state clearly and might even overstate in our previous manuscript. DFT-D calculated gas binding energies are usually in reasonable, qualitative agreement with experimental heat of adsorptions, especially when involves structural transformations during gas sorption. Typically, their differences can be ~ 20% (J. Phys. Chem. A 2017, 121, 4139–4151). On the other hand, the binding configurations from DFT calculations were illustrated only with the major H···F interactions while the distances of other weaker intermolecular interactions cannot be well identified for quantitative comparison. We realized the comparison of these interactions and binding energy is inappropriate, which can only be performed qualitatively. Accordingly, we have revised our claim on DFT modeling results by changing the relevant comparison in the revised manuscript.

Manuscript: Page 9

“...H···F hydrogen-bond interactions (1.85–2.36 Å, Figures 3c and S34). The static binding energy of SIFSIX-dps-Cu for C₂H₂ is calculated to be 60.3 and 67.5 kJ mol⁻¹, respectively...”

Comment 11. *The Clausius-Clapeyron plot used to calculate the Q_{st} values should be presented.*

Author Response: Thanks for the comment. The Clausius-Clapeyron plots for Q_{st} values have been added in the revised supporting information as Figure S36.

Modifications:

Supplementary information:

Figure S36. Clausius-Clapeyron plots and corresponding Q_{st} of (a) SIFSIX-dps-Cu, (b) GeFSIX-dps-Cu, and (c) NbOFFIVE-dps-Cu.

Comment 12. There is a “roll-up” of CO_2 in breakthrough curves on SIFSIX-dps-Cu, and it becomes more obvious with the increasing of flow rate, please explain it.

Author Response: Thanks for the comment. The “roll-up” effect can be attributed to the displacement of minor co-adsorbed CO_2 by C_2H_2 . With the increase of flow rate, the gas displacement was performed more rapidly, resulting in a more significant “roll-up” in the curves.

Modifications:

Manuscript, Page 11:

“At 5 and 10 mL min⁻¹, the roll-up phenomenon becomes significant that might induce by faster gas displacement at higher flow rate (Figure 4b).”

Comment 13. The C_2H_2 isotherms at 298 K of NbOFFIVE-dps-Cu in this work and ZUL-220 (Shen, J., He, X., Ke, T. et al. Nat Commun 11, 6259 (2020)) is very different. Is it possible that the water molecules is still in the framework? As the NbOFFIVE-dps-Cu sample is just washed by the methanol and evacuated at room temperature.

Author Response: Thanks for the comment. ZUL-220 in literature (Ref. 42: Nat Commun 11, 6259 (2020)) was reported to activate at room temperature under vacuum, the same as that for NbOFFIVE-dps-Cu in this work. Both MOFs showed stepwise sorption isotherms for CO₂ at 195 K with almost identical adsorption behavior, including gate-opening at $P/P_0 \sim 0.2\text{--}0.3$ and a small-pore stage with the volume of about $0.09 \text{ cm}^3 \text{ g}^{-1}$. The total pore volume of NbOFFIVE-dps-Cu reaches to $0.30 \text{ cm}^3 \text{ g}^{-1}$ after pore-opening, higher than those of SIFSIX-dps-Cu ($0.20 \text{ cm}^3 \text{ g}^{-1}$) and GeFSIX-dps-Cu ($0.19 \text{ cm}^3 \text{ g}^{-1}$). To confirm the activated condition for C₂H₂ sorption at room temperature, we have further carried out sorption experiments upon activation at elevated temperatures. After activation at 313 and 333 K, the C₂H₂ sorption isotherms are basically unchanged as compared to the one activated at 298 K (Figure R1). As for different C₂H₂ isotherms compared to ZUL-220, sample morphology and particle size might account for the difference since their synthesis was performed with different salts. For clarity, we have added a comparison on C₂H₂ sorption in the revised manuscript.

Figure R1. Room-temperature C₂H₂ sorption isotherms of NbOFFIVE-dps-Cu at different activation temperatures.

Modifications:

Manuscript, Page 6:

“The C₂H₂ uptake of... NbOFFIVE-dps-Cu was measured to be... 1.65 mmol g^{-1} at 298 K and 1.0 bar, respectively (Figures 2a). These values match well the calculated capacities, which are... 1.48 mmol g^{-1} for ... NbOFFIVE-dps-Cu, respectively.”

Comment 14. In the synthesis of NbOFFIVE-dps-Cu, $(\text{NH}_4)_2\text{NbF}_6$ was used. Does $(\text{NH}_4)_2\text{NbF}_6$ react with $\text{Cu}(\text{BF}_4)_2 \cdot x\text{H}_2\text{O}$ to produce CuNbOF_5 ? Please check it.

Author Response: Thanks for the comment. During the synthesis process, NbF_6^{2-} can undergo hydrolysis to form NbOF_5^- , and using NbF_6^{2-} and Cu^{2+} as starting material has been reported to successfully synthesize NbOFFIVE-dps-Cu (Ref. 35: *Angew. Chem. Int. Ed.* **2020**, *59*, 3423–3428). The FT-IR spectra demonstrated the formation of $\text{Nb}=\text{O}$ by using KNbF_6 and $\text{Cu}(\text{BF}_4)_2 \cdot x\text{H}_2\text{O}$ for synthesis (Figure R2).

Figure R2. FT-IR spectra of ZU-62 synthesized by two methods (*Angew. Chem. Int. Ed.* **2020**, *59*, 3423–3428).

Comment 15. “Binary mixture ($\text{C}_2\text{H}_2/\text{CO}_2$, 50/50, v/v) were injected into a packed column with a flow rate of 2.0 mL min^{-1} and the clean separations of $\text{C}_2\text{H}_2/\text{CO}_2$ mixtures were realized by all dynamic layered materials (Figure 27).” The Figure 27 should be Figure S28.

Author Response: Thanks for your reminder. The error has been corrected in the revised manuscript.

Reviewer #3 (Remarks to the Author):

Overall Comment. In this manuscript, the authors report the enhancement of adsorption capacity and selectivity of acetylene in SIFSIX-dps-M MOFs by changing the metal (M). Among three isoreticular compounds investigated, SIFSIX-dps-Cu (a structure previously reported by the same authors in *J. Am. Chem. Soc.* **2020**, *142*, 9744)

has reached an uptake of 4.57 mmol/g for C₂H₂ at 298 K and 1 bar and the highest IAST selectivity of 1787 for C₂H₂/CO₂. The authors also analyzed host-guest interactions using dispersion-corrected density functional theory and grand canonical Monte Carlo simulations. The following questions should be addressed.

Author Response: We thank this reviewer for the valuable and constructive comments.

Comment 1. *There is too little data on stability test. Putting the sample in air for 24 h is not enough because it did not offer the exact humidity which is very important to the stability of material. The test should be done for a longer time (e.g. one week) under high humidity to prove the high stability. I suggest to add PXRD results after putting samples under high humidity, being soaked in water at different temperature and PH, and being soaked in other common solvents.*

Author Response: Thanks for your suggestion. According to your suggestion, PXRD and sorption studies of SIFSIX-dps-Cu upon various conditions mentioned in your comment (Figure S39). The results show that SIFSIX-dps-Cu is stable in water, several organic solvents, or under moisture for at least 7 days. The results and discussions have been added to the revised manuscript and supporting information.

Modifications:

Manuscript: Page 12

“Furthermore, PXRD and sorption studies upon various conditions show that SIFSIX-dps-Cu can maintain its crystallinity in water, several organic solvents or under moisture for at least 7 days (Figure S39).”

Supplementary information:

Figure S39. Stability study of SIFSIX-dps-Cu. (a) Powder X-ray diffraction patterns upon various conditions, and (b) corresponding CO₂ sorption isotherms at 195 K.

Comment 2. On page 5, the authors state that “... the activated crystal structures are determined by Rietveld refinements (Figures 1 and S7). Figure S7 shows PXRD patterns but the refinement details and refined structure data are not provided. These should be added in the supplementary information.

Author Response: Thanks for your reminder. According to your suggestion, the refinement details (Figure S7, caption) and refined structure data (Table S2-5) have been added to the revised supplementary information.

Modifications:

Supporting information, Figure S7:

“Rietveld structural refinement was performed on the **PXRD** data using the **GSAS** package. Due to the large number of atoms in the crystal unit cell, the ligand molecule and the gas molecule were both treated as rigid bodies during the Rietveld refinement, with the molecule orientation and center of mass freely refined. Final refinement on the positions/orientations of the rigid bodies, thermal factors, occupancies, lattice parameters, background, and profiles converge with satisfactory R factors.”

Table S2. Modelling study and refinement parameters of C₂H₂-loaded MOFs.

Table S3. Coordinates of the non-hydrogen atoms in C₂H₂-loaded SIFSIX-dps-Cu.

Table S4. Coordinates of the non-hydrogen atoms in C₂H₂-loaded GeFSIX-dps-Cu.

Table S5. Coordinates of the non-hydrogen atoms in C₂H₂-loaded NbOFFIVE-dps-Cu.

Comment 3. UTSA-300-Zn possesses a very high C₂H₂/CO₂ IAST selectivity of 743 (*J. Am. Chem. Soc.* 2017, 139, 8022), higher than other materials in Figure 2e and 2f except SIFSIX-dps-Cu. However, the author did not refer this material in the Figure 2e and Figure 2f. I think the author should add UTSA-300-Zn into these two figures. Also, the conditions (e.g. temp and pressure) in Figure 2e and 2f should be specified.

Author Response: Thanks for the comment. The performance of UTSA-300-Zn has been added to the comparison of Figures 2e and 2f in the revised manuscript. The measurement condition has also been specified in Figures 2e and 2f accordingly.

Modifications:

Manuscript: Figure 2

Figure 2. e) comparison of 50/50 C₂H₂/CO₂ IAST selectivity with representative MOFs; **f)** comparison plot between C₂H₂/CO₂ selectivity and C₂H₂ capacity with representative MOFs at 298 K under 1 bar.

Comment 4. The acetylene binding modes analysis is not new. It was already reported in the previous work (*J. Am. Chem. Soc.* 2020, 142, 9744). However, the DFT-D calculated binding modes (Site I and Site II) show very different F...H distances in the current paper (Figure 3c) and previous JACS paper (Figure 4d). Please explain in the revised paper.

Author Response: Thanks for your comment. We didn't state clearly in our previous manuscript. The $\text{CH}\cdots\text{F}$ interactions with distances of 1.85–2.36 Å in current Figure 3c were given by DFT-D calculation, whereas the ones (1.71–1.86 Å) in the previous JACS paper (Figure 4d in Ref. 39 J. Am. Chem. Soc. 2020, 142, 9744) were given by Rietveld refinements results of PXRD data. The C_2H_2 binding modes in both works are the same. For clarity, we have revised the figure caption of Figure 3.

Modifications:

Manuscript: Figure 3

Figure 3. Computational simulations for the density distribution of C_2H_2 on SIFSIX-dps-Cu a) at 1 kPa and b) at 100 kPa and 298 K; c) DFT-D calculated C_2H_2 binding mode in SIFSIX-dps-Cu; d) packing mode of C_2H_2 -loaded SIFSIX-dps-Cu structure.

Comment 5. The CO_2 breakthrough curves on SIFSIX-dps-Cu (Figure 4a) is much complicated than that of GeFSIX-dps-Cu and NbOFFIVE-dps-Cu. Can you explain the two steps before 15 min/g and the further roll up at 45 min/g in the breakthrough curves of CO_2 on SIFSIX-dps-Cu?

Author Response: Thanks for your comment. For the little non-monotonical change on breakthrough profile of SIFSIX-dps-Cu at around 15 min/g, it can be attributed to

the minor co-adsorption of CO₂ from the 50/50 C₂H₂/CO₂ mixture as indicated by its single-component adsorption isotherm for CO₂ (uptake of ~0.3 mmol/g at 0.5 bar). Accordingly, the small roll-up of the breakthrough curve for CO₂ at 45 min/g can be attributed to the displacement of minor co-adsorbed CO₂ by C₂H₂. In contrast, there was no co-adsorption of CO₂ in GeFSIX-dps-Cu during either single-component gas sorption or transient breakthrough adsorption process, which leads to a relatively smooth concentration curve. That phenomenon is consistent with the difference of structural changes in three MOFs upon C₂H₂-gas loading, where SIFSIX-dps-Cu shows a larger rotation on pyridinyl rings. We have added these discussions to the revised main text.

Modifications:

Manuscript: Page 10

“For small roll-up of the breakthrough curves for CO₂ in SIFSIX-dps-Cu, it can be attributed to the desorption of CO₂ induced by C₂H₂, which indicates a minor co-adsorption of CO₂ during the dynamic capture process, and finally displaced by C₂H₂. That phenomenon is consistent with the minor CO₂ uptake from single-component sorption isotherms.”

Comment 6. The uptake amount of C₂H₂ can be obtained from the simulation studies. This value should be compared with the experimental C₂H₂ uptake and discussed in the paper. Also, the sentence “Figure 4d shows the optimized structure of C₂H₂-loaded SIXSIX-dps-Cu sample ...” on page 9 has an error. “Figure 4d” should be “Figure 3d”.

Author Response: Thank you for your suggestion. According to the simulation results, the calculated uptakes of SIFSIX-dps-Cu, GeFSIX-dps-Cu, and NbOFIVE-dps-Cu for C₂H₂ are 4.40, 3.71, and 1.48 mmol g⁻¹, respectively, which are comparable to their experimental uptakes. This result has been added to the revised main text. The error on figure citation in Page 9 has been corrected.

Modifications:

Manuscript: Page 9

“The C₂H₂ uptake of SIFSIX-dps-Cu, GeFSIX-dps-Cu, and NbOFIVE-dps-Cu were also evaluated by GCMC simulation showing capacity of 4.40, 3.71, and 1.48 mmol g⁻¹, respectively, which are comparable to their experimental uptakes.”

Comment 7. Compare Figure S7 with Figure S15, the PXRD peaks changed after loading with C₂H₂, revealing gate-opening of the pore. A more detailed explanation should be given to relate the changes in the peak position and in the structure, namely how the structure is changed?

Author Response: Thank you for your suggestion. The changes of PXRD patterns and corresponding structural transformation in the three isorecticular MOFs upon C₂H₂-loading are the same as the prototypical zinc analogue UTSA-300 (Ref. 13). For the PXRD patterns of the three MOFs here, the dramatic structural transformation results in different diffraction peaks with significantly different indices, showing major changes in the peaks positions at around 10°, 16–17° and 21–23°, which might be assigned to the sliding and expanding of 2D coordination layers. Upon C₂H₂-loading, the MOFs underwent a structural expansion to the open phase. These structural transformation processes were accompanied by ligand rotations and pore-opening (Figure S15), also revealed by UTSA-300 in Ref. 13. Accordingly, these discussions have been added to the revised manuscript and SI.

Modifications:

Manuscript: Page 6

“The changes of PXRD patterns and corresponding structural transformation in the three isorecticular MOFs upon desolvation or C₂H₂-loading are the same as the prototypical zinc analogue UTSA-300 (Figures S2-S4 and Tables S2-S5)¹³.”

Supplementary information: Figure S15

Figure S15. The active and C₂H₂-loaded structures of (a) NbOFFIVE-dps-Cu, (b) GeFSIX-dps-Cu, (c) SIFSIX-dps-Cu, and (d) UTSA-300. The torsion angle was measured from C–N bond on pyridinyl ring to Cu–F/O bond.

Supplementary information: Figure S16

Figure S16. Powder X-ray diffraction patterns for Rietveld refinement of C_2H_2 -loaded (a) SIFSIX-dps-Cu, (b) GeFSIX-dps-Cu, and (c) NbOFFIVE-dps-Cu. The C_2H_2 accumulation patterns at Site I and Site II of C_2H_2 -loaded (d,g) SIFSIX-dps-Cu, (e,h) GeFSIX-dps-Cu, and (f,i) NbOFFIVE-dps-Cu.

Comment 8. In Figure S9, the BET surface areas are estimated based on CO_2 sorption data. I wonder if the result for NbOFFIVE-dps-Cu is accurate enough. How are the points chosen from the isotherm for fitting with such a shape? What is the pore size distribution? Is it consistent with the pore size value shown in Figure 1?

Author Response: Thanks for your comment. Accordingly, the BET surface area plot of NbOFFIVE-dps-Cu has been added as Figure S10c in the revised SI, with good fitting indices. Enough points (7 points) were applied for the fitting, matching well with the BET model. It should be mentioned that the standard DFT pore size distribution from the 77 N_2 isotherms is not applicable for the ultramicroporous NbOFFIVE-dps-

Cu here since the molecular size of N₂ is larger than the pore size, showing no N₂ uptake at 77 K. Alternatively, a rough pore size distribution (6.7 Å) was obtained from 195 K CO₂ sorption based on the Horvath-Kawazoe (H-K) model (Figure R3), which can be assigned to the pore cavities in this MOF (Site I, 3.0 × 4.5 × 4.2 Å³; Site II, 4.0 × 4.5 × 4.2 Å³, Figure S5). But it should be noted that result might be of significant deviation because CO₂ molecules at 195 K are far different from the ideal gas model, especially in such ultramicropore. The size values in Figure 1 are for pore aperture rather than pore cavity. For clarity, we have revised the figure caption of Figure 1.

Figure R3. The pore size distribution of NbOFFIVE-dps-Cu from 195 K CO₂ sorption.

Modifications:

Manuscript: Figure 1

“...with varying pore aperture size”

Supplementary information:

Figure S10c. BET calculation based on CO₂ adsorption isotherm of (c) NbOFFIVE-dps-Cu at 195 K.

Comment 9. Page 13. The authors state that “Single crystal X-ray diffraction data for SIFSIX-dps-Cu, GeFSIX-dps-Cu, and NbOFFIVE-dps-Cu were collected at 193(2) K ...”. Since the single crystal data of SIFSIX-dps-Cu has already been reported by the authors before, why is there a need to correct data again? If the data should in Table 2 are from the JACS 2020 paper, then it should be so indicated in the table, and SIFSIX-dps-Cu should be removed from the above statement.

Author Response: Thanks for your reminder. According to your suggestion, the statement has been revised to “GeFSIX-dps-Cu and NbOFFIVE-dps-Cu”. We have also indicated the 2020 JACS paper (Ref. 39) in Table S1.

Modification:

Manuscript: Page 14

“Single crystal X-ray diffraction data for GeFSIX-dps-Cu and NbOFFIVE-dps-Cu were collected at 193(2) K...”

Comment 10. None of Supporting Tables are mentioned in the main text. They need to be cited.

Author Response: Thanks for your reminder. All the supplementary information including figures and tables have been cited in the revised main text.

Comment 11. What does “transit breakthrough experiments” mean?

Author Response: Thanks for your comment. We have corrected the term “Transit breakthrough experiments” to “Transient breakthrough experiments” that is time-dependent gas adsorption from dynamic mixture flow in fixed-bed adsorbent.

Modifications:

Manuscript:

“**Transient** breakthrough experiments.”

REVIEWER COMMENTS

Reviewer #1 (Remarks to the Author):

All of the concerns have been well addressed.

Reviewer #2 (Remarks to the Author):

In this revised manuscript, the authors have well addressed my concerns and now I can suggest publishing it on current form, no further change needed.

Reviewer #3 (Remarks to the Author):

The authors have made great effort to address the reviewers' questions by carrying out various additional experiments. I am satisfied with their answers to most of my comments except a few minor issues as given below.

1. Comment #1. Regarding moisture stability tests, the authors included PXRD and CO adsorption data in Figure S39. Please give a quantitative measure of the humidity used in the experiments, namely actual relative humidity (RH) value.

2. Comment #5. If GeFSIX-dps-Cu does not adsorb CO₂ but SIFSIX-dps-Cu co-adsorbs CO₂ and C₂H₂, wouldn't the breakthrough time for GeFSIX-dps-Cu is shorter than that of SIFSIX-dps-Cu? But in Fig. 4a, they are nearly the same. Can this be explained? In addition, based on the CO₂ isotherm of NbOFFIVE-dps-Cu, the CO₂ uptake is obviously higher than that of SIFSIX-dps-Cu at 298 K and 0.5 bar, however, in Fig. 4a, the CO₂ breakthrough time of NbOFFIVE-dps-Cu is also about the same as that of SIFSIX-dps-Cu. Why is this so?

Author Responses to Reviewers' Comments

Reviewer #3 (Remarks to the Author):

Overall Comment. The authors have made great effort to address the reviewers' questions by carrying out various additional experiments. I am satisfied with their answers to most of my comments except a few minor issues as given below.

Author response: We thank Reviewer #3 for the valuable comments and constructive suggestions.

Comment 1. Regarding moisture stability tests, the authors included PXRD and CO adsorption data in Figure S39. Please give a quantitative measure of the humidity used in the experiments, namely actual relative humidity (RH) value.

Author response: Thank you for your reminder. Accordingly, we have clarified the relative humidity (RH 53%) in Figure S39, which was controlled by saturated $\text{Mg}(\text{NO}_3)_2$ solution. The detail has also been added to figure caption of Figure S39 in the revised supporting information.

Modifications:

Supplementary information:

Figure S39. Stability study of SIFSIX-dps-Cu. (a) Powder X-ray diffraction patterns upon various conditions, and (b) corresponding CO₂ sorption isotherms at 195 K. **Note:** The humidity is controlled by saturated $\text{Mg}(\text{NO}_3)_2$ solution at 298 K and 1.0 bar, with a relative humidity of RH 53%.

Comment 5. If GeFSIX-dps-Cu does not adsorb CO₂ but SIFSIX-dps-Cu co-adsorbs CO₂ and C₂H₂, wouldn't the breakthrough time for GeFSIX-dps-Cu is shorter than that of SIFSIX-dps-Cu? But in Fig. 4a, they are nearly the same. Can this be explained? In addition, based on the CO₂ isotherm of NbOFFIVE-dps-Cu, the CO₂ uptake is obviously higher than that of SIFSIX-dps-Cu at 298 K and 0.5 bar, however, in Fig. 4a, the CO₂ breakthrough time of NbOFFIVE-dps-Cu is also about the same as that of SIFSIX-dps-Cu. Why is this so?

Author Response: Thank you very much for clarifying and pointing this out. We agree that the breakthrough time of dynamic gas flow in porous material is basically proportion to their equilibrium uptake capacity during static and single-component gas sorption. It is true that the equilibrium CO₂ uptake of NbOFFIVE-dps-Cu (0.95 mmol g⁻¹) is higher than that of SIFSIX-dps-Cu (0.36 mmol g⁻¹) at 298 K and 0.5 bar (Figure R1), while the latter is comparable with that of GeFSIX-dps-Cu (0.25 mmol g⁻¹) under same condition. It is thus understandable that the breakthrough time for CO₂ in SIFSIX-dps-Cu and GeFSIX-dps-Cu are nearly the same. As for the similarity between SIFSIX-dps-Cu and NbOFFIVE-dps-Cu on breakthrough time for CO₂, it is noted that the mixed-components sorption isotherm of NbOFFIVE-dps-Cu for C₂H₂/CO₂ (50/50, mol/mol) is almost identical to its single-component result for C₂H₂ (Figure S17), implying negligible CO₂ co-adsorption in this MOF for C₂H₂/CO₂ mixture. On the other hand, the adsorption heat of NbOFFIVE-dps-Cu for C₂H₂ (53.6 kJ mol⁻¹) is far higher than that for CO₂ (28.8 kJ mol⁻¹). It is thus speculated that C₂H₂ molecules would be preferentially adsorbed from dynamic C₂H₂/CO₂ mixture during breakthrough experiment, and its occupation on the adsorption site would exclusively block further adsorption of CO₂. For clarity, we have added this discussion to relevant section in the revised manuscript.

Figure R1 (for review only). CO_2 adsorption isotherms of SiFSIX-dps-Cu, GeFSIX-dps-Cu, and NbOFFIVE-dps-Cu at 298 K.

Modifications:

Manuscript: Page 11

“For the minor inconsistency between the CO_2 breakthrough curve and equilibrium CO_2 isotherm of NbOFFIVE-dps-Cu, it can be attributed to the preferential capture of C_2H_2 from the dynamic gas flow by this MOF that inhibits CO_2 adsorption, as indicated by distinctly different adsorption heats (C_2H_2 : 53.6 kJ mol^{-1} , CO_2 : 28.8 kJ mol^{-1}) and dual-components adsorption result (Figure S17).”

REVIEWER COMMENTS

Reviewer #1 (Remarks to the Author):

I have carefully looked at the reviewer #3's concerns. The first concern was well answered.

The authors' reply to the second concern seems to be reasonable, but not in satisfactory way. The authors mentioned that the adsorption heat for CO₂ is 28.8 kJ/mol, much smaller than that of 53.6 kJ/mol for C₂H₂ using NbOFFIVE-dps-Cu. However, this fact does not help to understand why the CO₂ breakthrough time using NbOFFIVE-dps-Cu is similar to that of SiFSIX-dps-Cu, of which the CO₂ binding energies (page 9 in manuscript) are reported to be 41.9 and 33.7 kJ/mol at site I and II (larger than the adsorption heat of 28.8 kJ/mol for CO₂).

Author Responses to Reviewers' Comments

Reviewer #1 (Remarks to the Author):

Comment 1. I have carefully looked at the reviewer #3's concerns. The first concern was well answered.

Author response: We thank this reviewer for all the valuable comments, which have significantly helped us to improve the quality of this work.

Comment 2. The authors' reply to the second concern seems to be reasonable, but not in satisfactory way. The authors mentioned that the adsorption heat for CO₂ is 28.8 kJ/mol, much smaller than that of 53.6 kJ/mol for C₂H₂ using NbOFFIVE-dps-Cu. However, this fact does not help to understand why the CO₂ breakthrough time using NbOFFIVE-dps-Cu is similar to that of SIFSIX-dps-Cu, of which the CO₂ binding energies (page 9 in manuscript) are reported to be 41.9 and 33.7 kJ/mol at site I and II (larger than the adsorption heat of 28.8 kJ/mol for CO₂).

Author response: Thank you for pointing this out. We have realized that the relevant statements on the binding energy in our previous manuscript might have misled the audience. The CO₂ binding energy in SIFSIX-dps-Cu was simplified without taking into account the adsorption kinetics. Additional experiments were conducted to measure the adsorption kinetics of C₂H₂ and CO₂ in SIFSIX-dps-Cu at 0.5 bar and 298 K. The new experimental data (Figure S39) showed that a rapid C₂H₂ saturation with a capacity of 3.72 mmol g⁻¹ at 0.5 bar and 298 K was observed while no significant CO₂ uptake for more than 50 minutes, which strongly suggests a molecular sieving or kinetic-controlled separation mechanism for the separation of C₂H₂ over CO₂ in SIFSIX-dps-Cu. Given that the inaccessibility of the pores in SIFSIX-dps-Cu to CO₂, the discussion on the binding energy for CO₂ seems inappropriate, and to a certain extent, might mislead the audience. Accordingly, we have deleted the relevant discussion on the calculated binding energy for CO₂ in SIFSIX-dps-Cu in the manuscript.

For NbOFFIVE-dps-Cu, both C₂H₂ and CO₂ can be adsorbed but with a significant

difference in their binding energy (C_2H_2 : 53.6 kJ mol^{-1} , CO_2 : 28.8 kJ mol^{-1}), the separation of which is equilibrium-controlled. Thus, C_2H_2 was preferentially adsorbed in dynamic $\text{C}_2\text{H}_2/\text{CO}_2$ breakthrough separation by occupying the accessible binding sites in NbOFFIVE-dps-Cu and thus blocking the co-adsorption of CO_2 . Corresponding desorption curves of NbOFFIVE-dps-Cu in the regeneration process after $\text{C}_2\text{H}_2/\text{CO}_2$ breakthrough have confirmed negligible CO_2 co-adsorption (Figure S40). Accordingly, these results and discussions have been updated in the revised manuscript and supporting information.

Modifications:

Manuscript: Page 11

“The C_2H_2 and CO_2 adsorption kinetics on SIFSIX-dps-Cu showed no significant CO_2 uptake under a sufficiently long period of time but rapid C_2H_2 saturation with a capacity of 3.72 mmol g^{-1} at 0.5 bar and 298 K, indicating that CO_2 cannot diffuse into the framework of SIFSIX-dps-Cu (Figure S39).”

Supplementary information:

Figure S39. The C_2H_2 and CO_2 adsorption kinetic curves of SIFSIX-dps-Cu at 0.5 bar and 298 K.

Manuscript: Page 11

“The corresponding desorption curves of NbOFFIVE-dps-Cu in the regeneration

process after C_2H_2/CO_2 breakthrough have also confirmed negligible CO_2 co-adsorption (Figure S40).”

Supplementary information:

Figure S40. The signals of desorbed gases from NbOFFIVE-dps-Cu in the regeneration process of breakthrough separation.

REVIEWER COMMENTS

Reviewer #1 (Remarks to the Author):

The kinetics-controlled mechanism nicely explain the phenomenon.

Author Responses to Reviewers' Comments

Reviewer #1 (Remarks to the Author):

Overall Comment. The kinetics-controlled mechanism nicely explain the phenomenon..

Author response: We thank Reviewer #1 for the support on the publication of this work.